# Mechanism of the noncatalytic oxidation of soot using in situ transmission electron microscopy

Ming Gao[1,2,3], Yongjun Jang[4], Lu Ding [1,3] ✉, Yunfei Gao[1,3], Sheng Dai [4] ✉, Zhenghua Dai[1,3], Guangsuo Yu[1,3], Wenming Yang [2] ✉ & Fuchen Wang [1,3] ✉

Soot generation is a major challenge in industries. The elimination of soot is particularly crucial to reduce pollutant emissions and boost carbon conversion. The mechanisms for soot oxidation are complex, with quantified models obtained under in situ conditions still missing. We prepare soot samples via noncatalytic partial oxidation of methane. Various oxidation models are established based on the results of in situ transmission electron microscopy experiments. A quantified maturity parameter is proposed and used to categorize the soot particles according to the nanostructure at various maturity levels, which in turn lead to different oxidation mechanisms. To tackle the challenges in the kinetic analysis of soot aggregates, a simplification model is proposed and soot oxidation rates are quantified. In addition, a special core-shell separation model is revealed through in situ analysis and kinetic studies. In this study, we obtain important quantified models for soot oxidation under in situ conditions.

Soot is a byproduct of incomplete combustion and contributes to pollution[1]. It is produced in large quantities not only in internal combustion engines[2] but also during many chemical industrial processes involving fossil fuels, such as entrained flow gasification of coal or biomass and noncatalytic partial oxidation (NCPOX) of gaseous hydrocarbons[3]. In industrial processes that utilize fossil fuels, the production of soot results in carbon loss and a low yield of syngas ($H_2$ and CO). Soot is easily entrained out by syngas and influences subsequent processes. Soot is thought to have a considerable influence on global warming[4], second to that of $CO_2$[5,6] and even greater than those of methane and other halocarbon greenhouse gases[7]. In addition, soot can be inhaled by humans and thus increases the risk of respiratory diseases[8]. Therefore, eliminating soot is quite important for environmental protection and the stable operation of industrial installations.

The most effective approach to eliminate soot is to oxidize it before emission. The oxidation of soot can be categorized into two types: high-temperature oxidation (>800 °C) and low-temperature oxidation (300 °C–700 °C)[9]. The oxidation rate is low at low temperatures, and it is necessary to use catalysts to effectively eliminate soot[10]. There have been sufficient studies on low-temperature oxidation[11–14], which were based on the exhaust systems of engines. However, temperatures in industrial furnaces typically exceed 800 °C. In this range, the reaction rate of oxidation is greater. The competition between the formation and consumption of soot is greater. Soot undergoes cracking to form fine particles by oxidation in the furnace. We even collected a large quantity of highly oxidized soot particles in the NCPOX industrial furnace[3]. Therefore, it is possible to eliminate soot before emission at high temperatures.

On the macroscopic scale, the oxidation reactivity of soot is correlated to its physicochemical properties. Extensive studies have shown that the oxidation reactivity of soot is related to its carbon structure[3,15,16]. The soot ageing process in high-temperature furnaces

[1]Institute of Clean Coal Technology, East China University of Science and Technology, Shanghai 200237, P.R. China. [2]Department of Mechanical Engineering, National University of Singapore, Singapore 117576, Singapore. [3]Engineering Research Center of Resource Utilization of Carbon-containing Waste with Low-carbon Emissions, Ministry of Education, Shanghai 200237, P.R. China. [4]Key Laboratory for Advanced Materials and Feringa Nobel Prize Scientist Joint Research Center, Institute of Fine Chemicals, School of Chemistry & Molecular Engineering, East China University of Science and Technology, Shanghai 200237, P.R. China. ✉e-mail: dinglu@ecust.edu.cn; shengdai@ecust.edu.cn; mpeywm@nus.edu.sg; wfch@ecust.edu.cn

involves carbonization, surface growth, and coagulation, resulting in the formation of soot particles with different degrees of graphitization[17]. More crystal layer defects, shorter lattice fringe lengths, and a less graphite-like structure endow soot with better reactivity and a lower oxidation activation energy[18,19]. However, these studies required estimating the oxidation mechanisms, which were not understood in detail.

On the microscopic scale, some empirical oxidation models, such as the shrinking core model[20], the homogeneous reaction model[21], and the random pore model[22,23], describe the oxidation processes of carbonaceous materials. However, the oxidation of soot is very complex because of the special particle nanostructures of soot. These nanostructures make oxidation process modeling quite challenging. For example, mature soot consists of an onion-like graphitized carbon shell and an amorphous carbon core[24]. Studies have reported the hollow nature of soot particles during oxidation[25-27], and but its evolution has not yet been revealed by in situ observations. Gao et al.[28] found that soot particles collected from an industrial furnace at high temperature had hollow nuclei. This finding indicated that soot presented a selective oxidation phenomenon in high-temperature furnaces. It is thus necessary to study the effect of nanostructures on soot oxidation.

In situ transmission electron microscopy (TEM) provides detailed real-space information with a high spatial resolution. It allows researchers to directly analyze a particle's real-time reactions at the atomic scale. During in situ TEM experiments in the gas phase, the differentially pumped environmental TEM approach (ETEM, open type) and the windowed gas cell approach (closed type)[29,30] are normally utilized. Sediako et al.[31] observed noncatalytic oxidation of soot by ETEM. Through observations on the soot particles, it was hypothesized that soot has different oxidation modes. Sediako et al.[32] and Toth et al.[33] conducted in situ observations under various pressures and temperatures. The surface reaction and densification during soot oxidation were observed, and soot with larger diameters was likely to undergo highly nonreactive surface oxidation. Naseri et al.[34] and Dadsetan et al.[35] further discovered that the electron beams of TEM also have impacts on carbon black particle conversions under in situ oxidation conditions using ETEM and scanning transmission electron microscopy (STEM). However, most studies have been limited to qualitative research on soot oxidation, and kinetic studies have not been conducted on oxidation modes of soot by in situ conditions.

Soot samples always consist of particles with different properties. The properties of soot produced under turbulent conditions can be more inhomogeneous because of the complex production environment in the furnace. These different particles always present various oxidation behaviors. Therefore, there is no basis to predict the oxidation mode of a soot sample. Most current characterization methods provide only the average properties of the soot samples. It is quite challenging to distinguish between different particles. To our knowledge, the mechanism and quantitative model of soot oxidation at high temperatures have rarely been studied due to the limitations of the in situ method. Therefore, it is necessary to establish an in situ strategy to obtain a reliable model to accurately describe the high-temperature oxidation behavior of soot.

In this study, a series of soot samples were produced at different molar ratios of $O_2$ to $CH_4$ ($O_2/CH_4 = 0.5$, 0.6, 0.7, and 0.8), which was denoted as S1, S2, S3, and S4. The nanostructure of soot was characterized. A maturity parameter was established to quantify the differences in the nanostructures of soot particles at various maturity levels. The oxidation behaviors of these soot particles were evaluated by in situ TEM. The results can be used in kinetics studies by establishing a variety of oxidation models. A simplification approach was proposed for oxidation models of soot aggregates. The relationship between the nanostructure and macroproperties was established by considering the maturity parameters and various characterization methods, including energy dispersive X-ray spectroscopy (EDS), Raman spectroscopy, high-resolution TEM (HRTEM), and thermogravimetric analysis. The application of oxidation models in real combustion modeling was discussed by proposing two approaches. This study developed good correlations between quantitative mathematical models and advanced in situ TEM observations for the characterizations of soot oxidation.

## Results and discussion

### Nanostructure of soot from high-temperature furnace

Solid products and liquid products were separated at higher temperatures. No soot precursors or liquid-like incipient soot (<10 nm) were observed in the TEM images. The collected particle size ranged from ~26 nm to ~391 nm. The morphology of the soot is shown in Supplementary Fig. S1. The average particle size, $\bar{D}_P$, decreased from 133.76 nm to 83.22 nm along with an increase in the $O_2/CH_4$ ratio from 0.5 to 0.8 during the production processes. Compared with the soot collected from the laminar flame, the soot from high-temperature furnace had a much larger average particle size. The effects of coalescence and surface growth were more significant in high-temperature furnace. The soot samples also presented a turbulent flame soot character. In the NCPOX process, the turbulent jet flame in the furnace was divided into a jet flow zone, a recirculation zone, and a reforming zone[36]. The different particle residence times in these zones resulted in an inhomogeneous distribution of the primary particle size of each soot sample.

The nanostructures of particles of different sizes were observed by HRTEM. According to the different nanostructures, soot particles were divided into 3 types, as shown in Fig. 1a: young soot, partially matured soot, and mature soot.

The young soot was in the early stage of ageing[37,38]. Soot transitions from coalescence to agglomeration and forms a particle structure from liquid-like incipient soot. Therefore, the particle size of young soot was larger than that of incipient soot. The particle boundaries were rough, and the interior showed a loose and multipored structure. The polycyclic aromatic hydrocarbon (PAH) crystalline layers presented irregular orientations. The most obvious characteristic was that there was no graphitic shell. The main particle size of the young soot ranged from 26 nm to 100 nm. The partially matured soot was in the middle stage of ageing[39]. Spherical structures had formed in the primary particles. This stage involves mainly dehydrogenation and surface reactions. The particles began to form graphitic shells, but the boundaries were still rough. The arrangement of PAH crystals changed to a concentric orientation. The main particle size was from 53 nm to 143 nm. The mature soot was in the late stages of ageing. The particles had formed smooth graphitic shells with a highly spherical shape. Mature soot formed larger aggregates at this stage. The main particle size was from 58 nm to 281 nm. The particle size distribution of the soot sample is shown in Fig. 1b.

The concentrations of the three types of soot are shown in Fig. 1c. The proportion of mature soot decreased with increasing $O_2/CH_4$ ratio. The proportion of partially matured soot peaked at $O_2/CH_4 = 0.7$. The proportion of young soot increased obviously from $O_2/CH_4 = 0.7$. The soot type of the samples was controlled by the $O_2/CH_4$ ratio. Researchers can produce soot of a particular structure to utilize or facilitate elimination.

### Maturity levels of soot particles with different nanostructures

There was a significant difference in the nanostructure of the three types of soot. However, the evolution of the nanostructure was the same for different soot samples. The three types of soot were at different maturity levels. A maturity parameter $M$ was established to quantify the differences in the nanostructure of soot particles.

Soot maturity mainly depends on two independent parameters, the C/H ratio and the primary particle size[40]. The C/H ratio reflects the

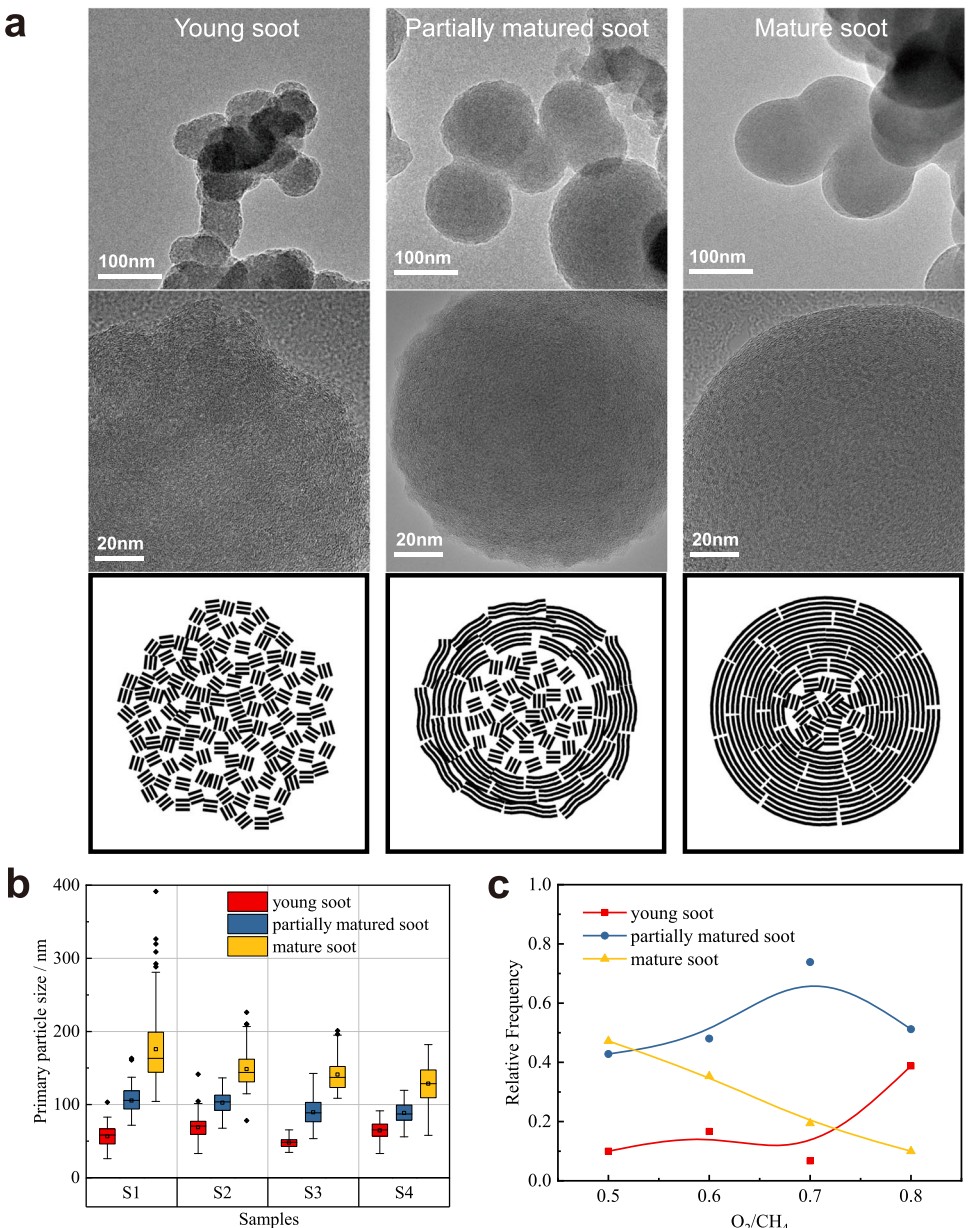

**Fig. 1 | The classification and distribution of soot particles in samples. a** The classification of soot particles was based on the nanostructure of soot particles. **b** The particle size distribution of three types of soot. There were overlaps in the size ranges of different types of particles. It was more accurate to assess the maturity of particles based on their structure than relying on particle size alone. **c** The maturity of soot samples became younger with the $O_2/CH_4$ increased. Source data are provided as a Source Data file.

degree of dehydrogenation and carbonization at high temperatures. The particle size reflects the degree of surface growth. However, the C/H ratio of individual particles is difficult to obtain. In addition, it is not possible to determine the maturity from the particle size alone. Studies have shown that particle growth and graphitization result in the development of a fine carbon-lattice structure during maturation[41–43]. Therefore, a parameter described by the nanostructure was adopted. The fringe length and C/H ratio can reflect the size increase and carbonization.

The carbon atom number of PAH molecules can be calculated by using Eq. (1)[44].

$$C_{\mathrm{PAH},i} \approx \left[ (L_{a,i} + 0.211)/0.193 \right]^2 \tag{1}$$

where $L_{a,i}$ is the fringe length of the $i_{\mathrm{th}}$ PAH determined by HRTEM. The carbon-to-hydrogen ratio of $\left( \frac{C}{H} \right)_{\mathrm{PAH},i}$ can be expressed as

shown in Eq. (2)[45].

$$\left( \frac{C}{H} \right)_{\mathrm{PAH},i} = \left( \frac{C_{\mathrm{PAH},i}}{6} \right)^{0.43} \tag{2}$$

We found that Eq. (2) was suitable for plane PAHs. Therefore, it is appropriate for the prediction of immature soot. During maturation, defects can cause PAHs to curve from a plane structure to a three-dimensional structure. The mature soot contains more curved PAHs, which have higher C/H ratios. Here, we established a dimensionless parameter, $\tau$, to correct the hydrogen number of curved PAHs. As shown in Fig. 2a, b, the H atoms are distributed only on the edge of the PAH. The actual number of H atoms is related to the length of the sides. In HRTEM, $\tau$ reflects the tortuosity of a fringe. As shown in Fig. 2c, $\tau$ can be defined as shown in Eq. (3), where $L_a$ is the fringe length and $D$ is the

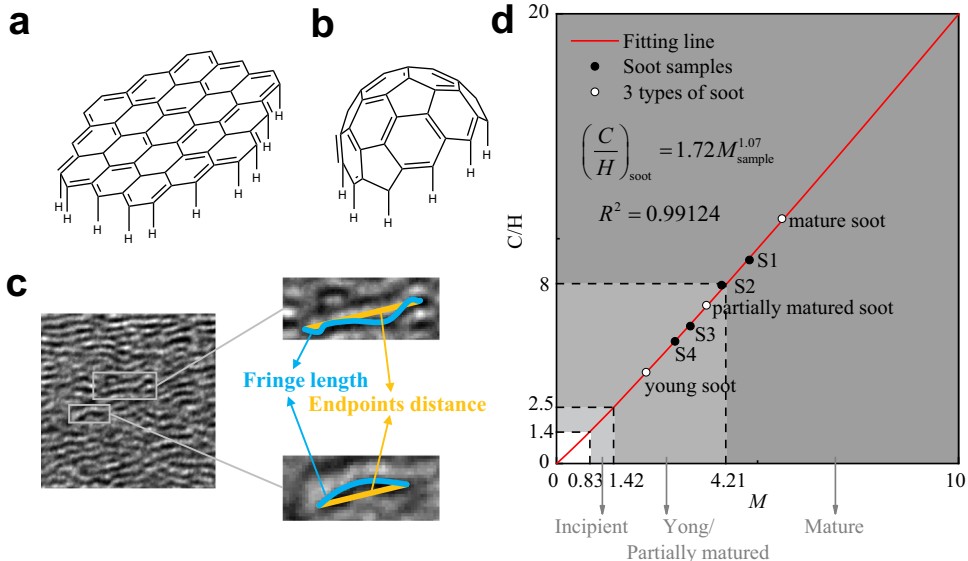

**Fig. 2 | The correction and the calculation result of maturity parameter.**
**a**, **b** When the plane PAHs curved to form three-dimensional structures, the number of hydrogen atom decreases further. **c** The tortuosity was defined as the ratio of fringe length to endpoints distance. **d** The maturity level of particles can be determined from the relationship between maturity parameter, $M$, and C/H ratios. Source data are provided as a Source Data file.

distance between the two endpoints.

$$\tau = \frac{L_a}{D} \qquad (3)$$

The corrected number of hydrogen atoms, $H_{PAH,i}$, can be expressed as shown in Eq. (4).

$$H_{PAH,i} = \frac{C_{PAH,i}}{\tau \left(\frac{C}{H}\right)_{PAH,i}} \qquad (4)$$

For a PAH with fewer than six rings, the molecule is too small to bend. Curved PAHs often form due to the five-membered carbon rings inside them. We selected corannulene ($C_{20}H_{10}$) as the first molecule to be curved. Corannulene is a PAH with nearly circular symmetry and 20 carbon atoms. When the carbon atom number of partially matured soot and mature soot is below 20, $\tau = 1$. Therefore, we defined the maturity $M_{particle,j}$ of a single particle $j$ as the ratio of the number of C atoms to the corrected number of H atoms of all PAHs, as shown in Eq. (5).

$$M_{particle,j} = \frac{\sum_i C_{PAH,i}}{\sum_i H_{PAH,i}} = \frac{\tau \sum_i (L_{a,i} + 0.211)^2}{2.404 \cdot \sum_i (L_{a,i} + 0.211)^{1.14}} \qquad (5)$$

When $L_{a,i} < 0.652$, $\tau = 1$; $L_{a,i} \geq 0.652$, $\tau = \frac{L_a}{D}$. The same method can be used to determine the average maturity of multiple particles, as shown in Eq. (6).

$$M_{sample} = \frac{\tau \sum_{i,j} (L_{a,i,j} + 0.211)^2}{2.404 \cdot \sum_{i,j} (L_{a,i,j} + 0.211)^{1.14}} \qquad (6)$$

When $L_{a,i,j} < 0.652$, $\tau = 1$; $L_{a,i,j} \geq 0.652$, $\tau = \frac{L_a}{D}$. The $M_{sample}$ values of the soot samples were determined and compared to the results of elemental analysis. The relationship is shown in Fig. 2d. Since the curve passes through the origin, the allometric function was performed.

$$\left(\frac{C}{H}\right)_{soot} = 1.72 M_{sample}^{1.07} \qquad (7)$$

Due to the different analytical principles of instruments, the macroscopic C/H ratio determined through Eq. (7) may be underestimated. The significance of $M_{sample}$ lies in reflecting the extent of growth and carbonization of the sample. It is generally recognized that the C/H ratio of incipient soot is 1.4-2.5 and that of mature soot is 8-20[46-49]. It can be determined from Eq. (7) that the $M_{sample}$ value of incipient soot is 0.83-1.42, and that of mature soot is 4.21-9.91. Most of the samples were found to be in the middle stage of maturation. This was attributed to the high concentration of partially matured soot in the samples. We correlated the 3 types of soot with the general sample by calculating the $M_{sample}$ value. The young soot corresponded to the soot samples with C/H ratios of 2.5-4.1, the partially matured soot corresponded to samples with C/H ratios of 4.1-8, and the mature soot corresponded to samples with C/H ratios or 8-20.

The calculated maturity level of soot samples well matched the corresponding nanostructure obtained by TEM, which further proved the rationality of $M_{sample}$. This enables the characterization of soot nanoparticles through quantitative analysis and correlates the nanostructures with macroproperties of soot. As the maturation process of soot is the same[50-52], the maturity level and corresponding oxidation model of soot from different conditions can be determined by $M_{sample}$.

## High-temperature oxidation processes of soot with different nanostructures

In situ oxidation experiments of soot particles were carried out. The oxidation modes depended greatly on the nanostructures of soot with different maturity levels.

The oxidation process of young soot is shown in Fig. 3a. The particles shrank and formed a core during oxidation. The primary particle size continued to decrease throughout the process. The reaction was very fast in the initial stage of oxidation. The particle size decreased, and the bond between particles was broken. The soot aggregate was divided into small particles. In the middle stage, the small particles formed many pore structures and cracked into pieces. The pieces agglomerated to form spherical clusters, which may be caused by van der Waals interactions[53]. In the final stage, the spherical clusters were continuously oxidized until a small amount of poorly reactive fragments remained. The in situ oxidation showed fragmentation and agglomeration during oxidation. This was related to the low

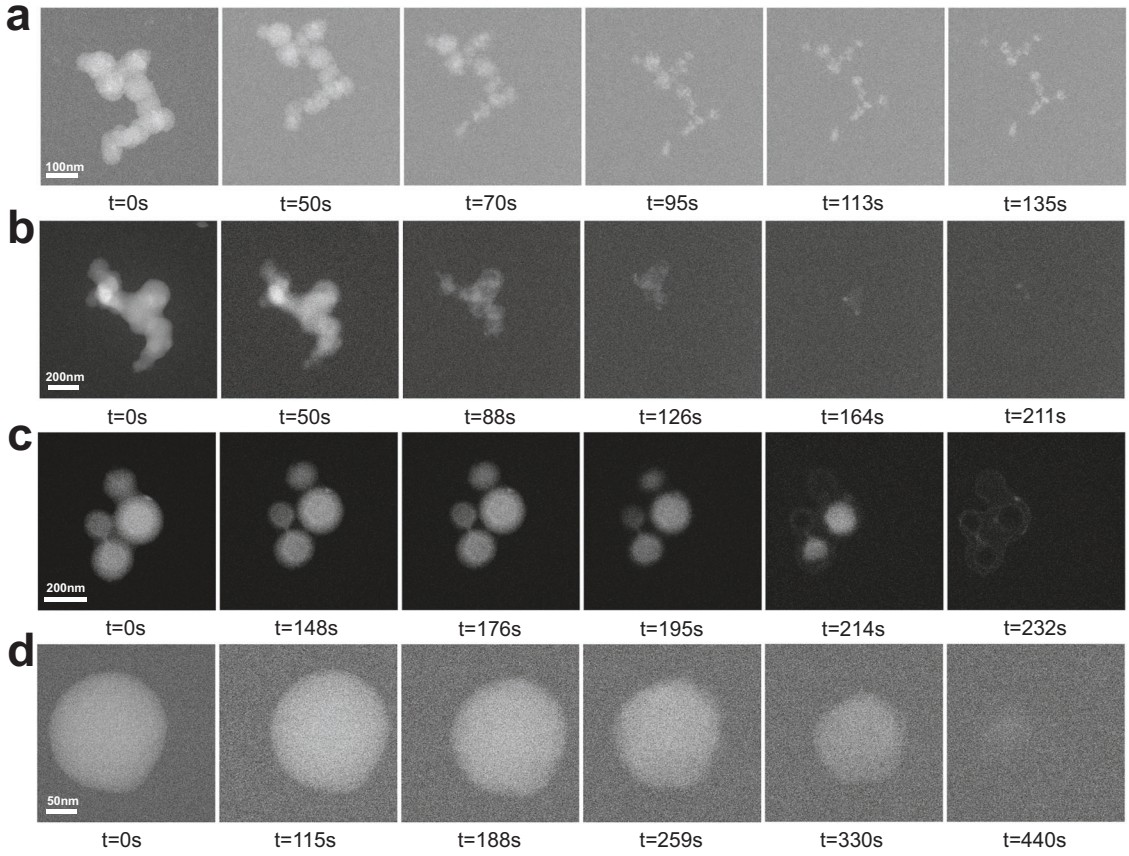

**Fig. 3 | Oxidation processes under in situ TEM. a** Young soot, **b**, **c** partially matured soot, and **d** mature soot.

carbonization degree of young soot. The oxidation of young soot was named the fast-shrinking core model (SCM).

The oxidation of partially matured soot occurred in two different modes. One is shown in Fig. 3b. The oxidation started from the interior of the particles, and the particle size was unchanged. Several hollow structures appeared inside the particles, which did not necessarily originate from the center of the primary particle. As the oxidation progressed, the hollow structures increased in size. The particles outside the aggregates were consumed earlier than the middle particles. The final stage was the oxidation of the carbon shell. When the hollow structures occupied most of the particle, the particles were not able to maintain the spherical structure and cracked into small pieces. These small pieces agglomerated and were oxidized.

The formation of hollow regions and the shell fragmentation of partially matured soot were evaluated by in situ TEM. The surface of partially matured soot is not dense enough, and oxygen easily entered the interior of the particles through pores. The reactivity of amorphous carbon inside the particles was much greater than that of the graphitized shell. Therefore, the interior was preferentially oxidized and formed an internal oxidation model (IOM).

Another kind of core-shell separation model (CSM) was observed by in situ TEM, as shown in Fig. 3c. In the early stage of the reaction, the reaction was very slow, and no apparent change was observed. As the reaction proceeded, the shell and the core were separated, and the core began to shrink. The core was always oxidized and remained spherical, and no hollow structures were observed in the core. The core did not always contract around the center of the particle sphere. Finally, a graphitized shell remained, which was difficult to oxidize. The particle size remained constant throughout the CSM oxidation process.

To study the CSM mechanism, the soot particle structure was further analyzed by HRTEM and EDS. Figure 4a shows the HRTEM image, HAADF-STEM image, and corresponding EDS elemental maps of C and O in particles subjected to the CSM. Figure 4b shows images of other soot particles. Some fine PAHs were observed in the shell part of the particles subjected to the CSM. The locations of these PAHs were consistent with the separated interfaces. These PAH structures caused carbon atoms to bond with sp³ or mixed sp²-sp³ hybridization, which induced bending and defects in the crystalline layers[54]. This is one possible reason for core-shell separation during oxidation. As shown in Fig. 4c, the shell of soot aggregated according to the CSM was held together by chemical bonds. The shell was formed by surface growth, and the separated interface was likely to be the initial position of surface growth. The oxidation reactivity of carbon on either side of the separated interface was significantly different. According to the EDS results, the distribution of C was uniform in the two kinds of particles. However, the concentration of O was much greater at the separated interface. These oxygen-containing functional groups can cause surface defects in graphite crystal layers. Furthermore, these functional groups reduced the reactivation energy for oxidation, which caused the carbon in the corresponding position to be preferentially consumed by oxidation[55,56]. This is another possible reason for the core-shell separation during the oxidation of partially matured soot.

CSM is different from the previously reported models, and the evolution mechanism of the CSM was revealed in this study. This provides researchers with a more comprehensive understanding of soot oxidation processes. The cage structure of soot formed by the CSM was stable and had strong oxidation resistance, which can be potentially used as a storage and transportation medium for drugs or protein[57].

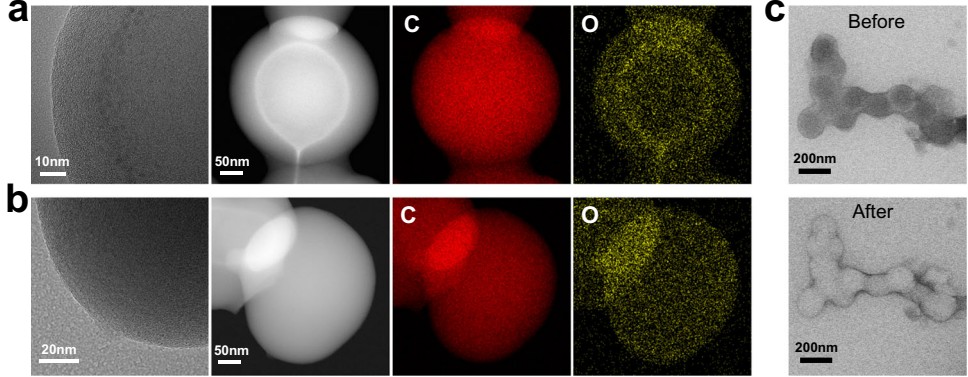

**Fig. 4 | The special structure of soot oxidized by CSM.** There are some fine PAHs locating at the separated interface of **a** soot particles of CSM, comparing to **b** other soot particles. Oxygen functional groups (yellow) are more concentrated at the separation interface, compared to carbon (red). **c** The graphitic shell of the soot remains intact and was not fragmented or oxidized.

The reaction time of mature soot was the longest among the oxidation models, as shown in Fig. 3d. The oxidation process started from the surface and proceeded only on the surface. The particle size decreased slowly. The periphery of the particles was damaged, resulted in the formation of several small-scale defects. However, no internal oxidation was observed in this case. It is indicated that the graphitized shell of mature soot is dense and thick enough. Naseri et al.[34] noted that the graphitic shell of soot was highly resilient to $O_2$, and the activation energy for surface oxidation was high. The mature soot presented an oxidation mode of slowly shrinking throughout the process with spherical particles. It was described as slow SCM.

The in situ oxidation showed that the oxidation modes were highly dependent on the nanostructure of the soot particles. The nanostructure changed with the maturation process, and the oxidation mode changed. At the same time, these nanostructures maintained the sphericity of the particles. As oxidation proceeded, these structures were destroyed, leading to fragmentation of the spherical particles.

## Mathematical models of soot oxidation

To accurately calculate the reaction rate of soot from the experimental results, mathematical models describing the oxidation process of soot were established. The intrinsic oxidation reaction rate was expressed as Eq. (8)[58] according to reaction (9)[59]. There was a first-order dependency on the oxygen concentration[60].

$$r = \frac{dx}{dt} = \lambda S_{soot} k(T) c_{O_2} \tag{8}$$

$$C_{soot} \cdot + O_2 \rightarrow C_{soot}^* \cdot + CO_2 \tag{9}$$

where $\lambda$ is the number density of the surface reactive site, $S_{soot}$ is the area of the reaction surface on the soot particle, $K(T)$ is the reaction rate coefficient at the reactive site related to temperature, and $c_{O_2}$ is the concentration of oxygen. The temperature term, $K(T)$, can be expressed as shown in Eq. (10) according to the Arrhenius equation.

$$k = Ae^{-E_a/RT} \tag{10}$$

$S_{soot}$ is related to the conversion rate $x$, and $x$ can be obtained by experiments. Therefore, accurate models must be established to calculate the $x$ values of different soot oxidation processes, and then kinetic parameters can be obtained. According to the reaction process of soot particles, it is assumed that the primary particles of soot are solid carbon spheres with uniform density. Based on the in situ oxidation process, the conversion rate $x$ of soot particles can be obtained from the projected area $S$.

The oxidation models of young soot and mature soot are shown in Fig. 5a, b. According to the volume formula for a sphere, the primary particle volume $V$ at a certain conversion and the initial volume $V_0$ can also be given by $S$ and $S_0$ as Eqs. (11) and (12):

$$V = \frac{4}{3} S \sqrt{\frac{S}{\pi}} \tag{11}$$

$$V_0 = \frac{4}{3} S_0 \sqrt{\frac{S_0}{\pi}} \tag{12}$$

where $S_0$ denotes the initial projected area. Since the particle density is uniform, $x$ can be obtained by using Eq. (13).

$$x = 1 - \frac{V}{V_0} = 1 - \frac{S^{\frac{3}{2}}}{S_0^{\frac{3}{2}}} \tag{13}$$

As shown in Fig. 5a, b, the surface area, $S_{soot}$, decreases as the oxidation progresses. Thus, the surface area of the particle can be expressed as a function of $x$.

$$S_{soot} = \frac{dV}{dR} = S_{soot,0} \cdot (1-x)^{\frac{2}{3}} \tag{14}$$

$$S_{soot,0} = S_{soot}\big|_{x=0} = \frac{dV}{dR}\bigg|_{S=S_0} = 4S_0 \tag{15}$$

where $S_{soot,0}$ is the surface area at $x = 0$, and $S_0$ is the initial projected area. Substituting Eqs. (13), (14), and (15) into Eq. (8) results in Eq. (16).

$$\frac{1}{S} d\left(1 - \frac{S^{\frac{3}{2}}}{S_0^{\frac{3}{2}}}\right) = 4\lambda A e^{-E_a/RT} c_{O_2} dt \tag{16}$$

Therefore, the activation energy of the oxidation reaction can be obtained.

For the IOM, it was assumed that hollow structures were formed at random positions inside the particles and that the particle size remained unchanged during oxidation. The hollow structure was regarded as spherical and grew from the nuclei to the boundary of the particle until the reaction was complete, as shown in Fig. 5c. Notably, the projection area measured during the experiment was the area of the annular part. The volume of the remaining unoxidized part is

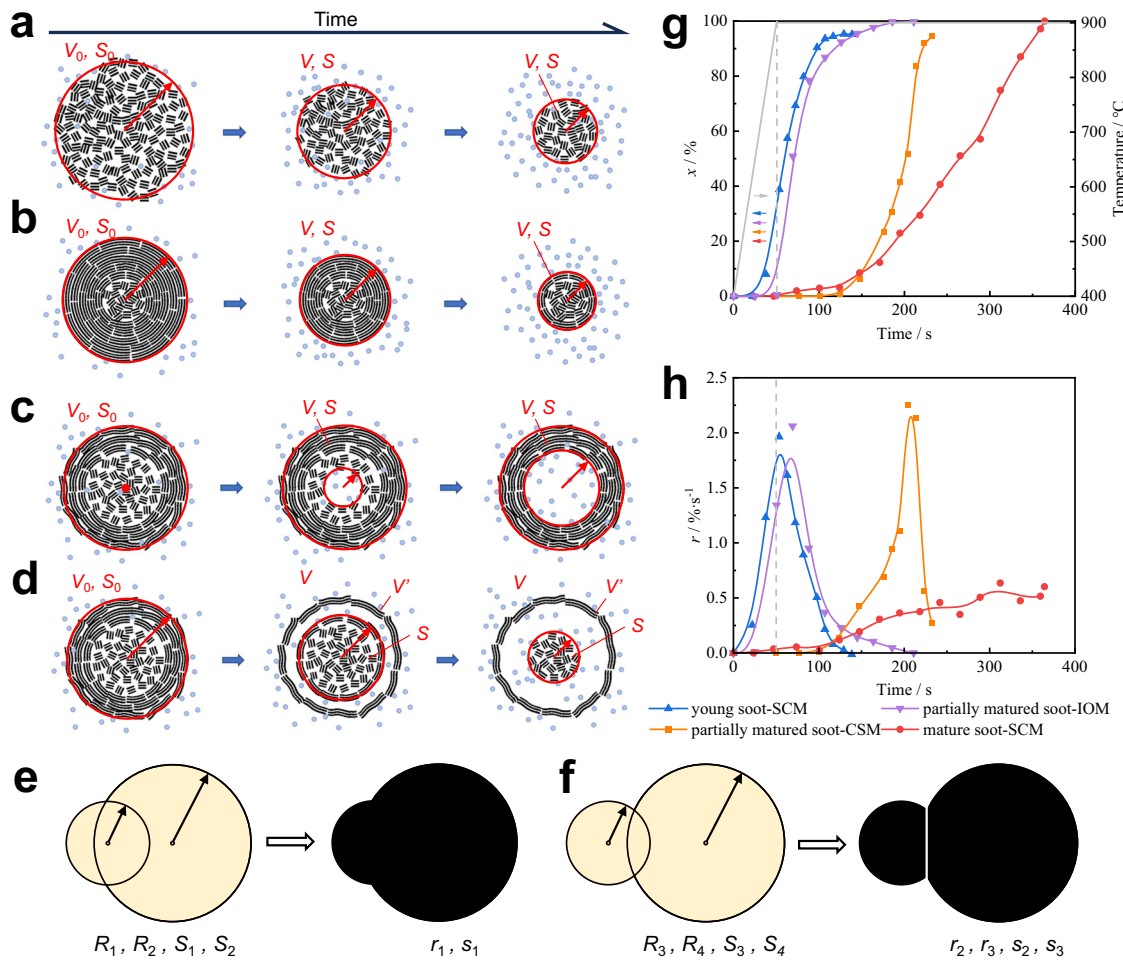

**Fig. 5 | Proposed models of different types of soot particles. a** SCM of young soot particles. **b** IOM of partially matured soot particles. **c** CSM of partially matured soot particles. **d** SCM of mature soot particles. **e, f** The simplification approach divided or merged the soot aggregates (yellow) into multiple adjacent equivalent spheres (black). The radius and projected area of the equivalent sphere were used for calculation as new parameters instead of the radius and projected area of soot aggregates. **g, h** The conversion rate and reaction rate of different particles were obtained by various simplified models. The reaction rate of models varied with time, which resulted in a challenge in the selection of models during the oxidation of multiple particles. Source data are provided as a Source Data file.

determined as shown in Eq. (17).

$$V = \frac{4}{3}\pi\left[\left(\frac{S_0}{\pi}\right)^{\frac{3}{2}} - \left(\frac{S_0 - S}{\pi}\right)^{\frac{3}{2}}\right] \quad (17)$$

According to Eq. (13), the expression of the conversion rate can be obtained by Eq. (18).

$$x = \frac{(S_0 - S)^{\frac{3}{2}}}{S_0^{\frac{3}{2}}} \quad (18)$$

For the IOM, the active area was the inner surface area of the particle, and the surface area increased gradually with the reaction progress. Thus, the surface area of the particle can be expressed as shown in Eq. (19).

$$S_{\text{soot}} = \frac{dV}{dR} = S_{\text{soot},0} \cdot x^{\frac{2}{3}} \quad (19)$$

The oxidation model of partially matured soot is shown in Fig. 5d. According to the in situ oxidation process in Fig. 3c, the contrast of the shell part was much lower than that of the core, indicating that the shell part was very thin. Assuming that the volume of the shell part is a certain value $V'$, the volume of the particle can be expressed as shown

in Eq. (20).

$$V = \frac{4}{3}\pi\left(\frac{S}{\pi}\right)^{\frac{3}{2}} + V' \quad (20)$$

Consequently, the conversion rate was found to be

$$x = 1 - \frac{V}{V_0} = \frac{S_0^{\frac{3}{2}} - S^{\frac{3}{2}}}{S_0^{\frac{3}{2}} + V''} \quad (21)$$

where

$$V'' = \frac{3\sqrt{\pi}}{4}V' \quad (22)$$

In the CSM, the reactivity of the residual shell was very poor; accordingly, the reactive area $S_{\text{soot}}$ was the surface area of the core.

$$S_{\text{soot}} = \frac{d(V - V')}{dR} = S_{\text{soot},0} \cdot (1-x)^{\frac{2}{3}} \quad (23)$$

The proposed models are suitable for soot samples with C/H ratios >2.5 and soot particles with $M_{\text{sample}}$ greater than 1.42. They may not be suitable for liquid-like incipient soot (<10 nm).

Soot is often found in aggregates. The fractal dimension of aggregates is much larger than that of primary particles, which leads to a large error of $x$ calculated by means of Eq. (13). Models for soot aggregates are needed.

Primary particles in TEM images can be identified by computer programming[61], which yields the equivalent radius ($R_1, R_2, R_3, \ldots\ldots$) and center coordinates ($x_1, x_2, x_3, \ldots\ldots y_1, y_2, y_3, \ldots\ldots$) of the primary particles in aggregates. Consequently, $x$ of soot aggregates can be obtained by these parameters. Nevertheless, the expressions for $x$ and $S_{soot}$ are too complicated, and they cannot be integrated by Eq. (8). The activation energy cannot be obtained by this approach. The complicated formula is unfriendly for engineering applications.

To address this issue, a simplified approach was established to obtain a suitable expression of $x$ for soot aggregates. The aggregates were processed as shown in Fig. 5e, f.

When the spherical center of the smaller primary particle was inside the larger particle, the two particles were treated as one spheroid (Fig. 5e). When the center of the smaller primary particle was outside the larger particle, the two particles were divided into two nonoverlapping spheroids at the intersection. Subsequently, a new equivalent spherical radius ($r_1, r_2, r_3, \ldots\ldots$) and projected area ($s_1, s_2, s_3, \ldots\ldots$) were obtained. The total projected area of the soot aggregate was the sum of the projected area of new spheroids, as shown in Eq. (24). Since the projected areas of new spheroids were calculated from the in situ TEM images, the total projected area was not affected by the simplification approach. The approach did not require calculating the volume of intersecting parts. The total volume was expressed as Eq. (25). Therefore, the conversion rate of soot aggregates $x$ was determined by means of Eq. (26).

$$S = \pi r_1^2 + \pi r_2^2 + \pi r_3^2 + \cdots = s_1 + s_2 + s_3 + \cdots \quad (24)$$

$$V = \frac{4}{3}\pi r_1^3 + \frac{4}{3}\pi r_2^3 + \frac{4}{3}\pi r_3^3 + \cdots = v_1 + v_2 + v_3 + \cdots \quad (25)$$

$$x = 1 - \frac{\sum s_i^{\frac{3}{2}}}{\sum s_{0i}^{\frac{3}{2}}} \quad (26)$$

The simplified aggregates no longer consisted of spherical particles but consisted of hypothetical equivalent spheres. The best advantage was that the conversion expression of aggregates was similar to that of single particles. This means that the model of aggregates is suitable for subsequent kinetic calculations. This simplified approach facilitated rapid and precise calculation of the reaction rates for various types of soot by the researchers.

Sometimes, small particles on the outside of the aggregate were eliminated first, which required researchers to track the oxidation progress of individual primary particles for accurate results. When the primary particle sizes in one aggregate were not very different, the projected area and volume were expressed as follows.

$$S = n\pi \bar{r}^2 \quad (27)$$

$$V = n\frac{4}{3}\pi \bar{r}^3 \quad (28)$$

where $n$ is the number of primary particles. Thus, Eq. (26) was reduced to Eq. (29), which is the same as Eq. (13).

$$x = 1 - \frac{S^{\frac{3}{2}}}{S_0^{\frac{3}{2}}} \quad (29)$$

In this situation, the in situ TEM results were used for kinetics studies.

The conversion rates of aggregates oxidized with the IOM and the CSM were expressed as Eq. (30) and Eq. (31) by the same approach. The simplification approach facilitates the kinetic calculations of soot aggregates oxidations, and these models could be used to quantify the oxidation process of soot.

$$x = \frac{\sum (s_{0i} - s_i)^{\frac{3}{2}}}{\sum s_{0i}^{\frac{3}{2}}} \quad (30)$$

$$x = \frac{\sum (s_0^{\frac{3}{2}} - s^{\frac{3}{2}})}{\sum (s_0^{\frac{3}{2}} + V'')} \quad (31)$$

The developed mathematical models were verified based on the experimental results obtained by in situ TEM. The relationships of the conversion rate and reaction rate changing with time are shown in Fig. 5g, h. The processed results accurately describe the soot oxidation behavior compared with the real-time oxidation processes. The first 50 seconds was the heating stage, and the reaction rate increased with increasing temperature. Young soot shows the highest reactivity. In the constant-temperature reaction stage, the reaction rate decreased gradually. This was related to the decrease in the active surface area. For the partially matured soot oxidized with the IOM, the reaction rate was very high in the early stage. This was due to the oxidation of amorphous carbon with high reactivity in the particle core. The reaction rate decreased gradually when the conversion rate exceeded 86.8%. This was related to the oxidation of the less active graphitic shell. For partially matured soot oxidized with the CSM, the reaction rate was very low in the early stage, which was due to shell separation. When the separation stage finished, the reaction rate began to increase, while the conversion rate was only 0.6%. As the oxidation proceeded, the core of the particle began to shrink. The reaction transitioned from the oxidation of a small amount of graphitized carbon to that of amorphous carbon. In the later stage, the reaction rate decreased rapidly when the conversion rate exceeded 83.8%. Finally, a highly graphitized shell with poor reactivity was left. Partially matured soot oxidized with CSM had the highest instantaneous reaction rate. The average reactivity of soot oxidized with the IOM and the CSM was similar for partially matured soot. The instantaneous reaction rate of mature soot was the lowest. The reaction rate of mature soot oxidized with SCM gradually increased with carbon conversion. The reaction rate eventually fluctuated around 0.5% s$^{-1}$. As the oxidation proceeded, the reactivity increased. The oxidation rate change during the processes reflected the nanostructure of soot particles.

The reaction rate varied greatly among different models, so it is very important to choose the correct oxidation model according to the maturity level of soot particles.

## Application of oxidation models

The oxidation reactivity of the four samples is shown in Fig. 6a. The reactivity, $R_{0.9}$, of soot increased as the ratio of $O_2/CH_4$ increased from 0.5 to 0.8 during the NCPOX process. The reactivity of soot depends greatly on the carbon structure[3]. The carbon structure of the samples was analyzed by Raman spectroscopy. With an increase in the $O_2/CH_4$ ratio from 0.5 to 0.8 during the NCPOX process, the integral band area ratio of the D band to G band ($I_{D1}/I_G$) of soot samples increased from 3.23 to 3.80, while the degree of graphitization of the particles decreased.

The relationship between $I_{D1}/I_G$ and $R_{0.9}$ is shown in Fig. 6b. The oxidation reactivity of soot was negatively correlated with $I_{D1}/I_G$. An increase in the maturity of the soot samples led to an increase in the average graphitization degree of the samples. This in turn resulted in a decrease in the oxidation reactivity. The quantified relationships between $O_2/CH_4$ ratio and maturity parameters are shown in Fig. 6c. According to the $M_{sample}$ of samples, S1 ($O_2/CH_4 = 0.5$) belongs to the

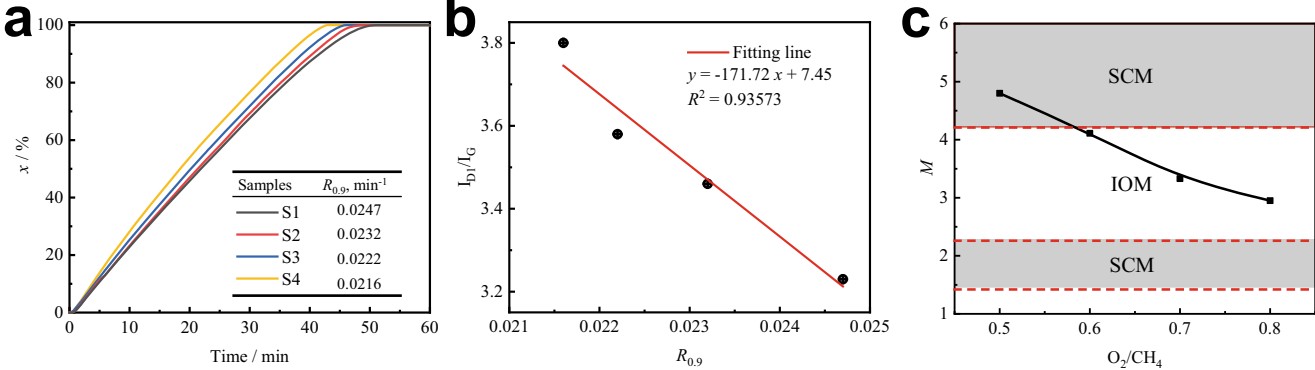

**Fig. 6 | Macroscopic properties of samples and corresponding oxidation models. a** The reactivity of soot increased as the ratio of $O_2/CH_4$ increased from 0.5 to 0.8. **b** The increase of reactivity was because the graphitization degree of soot samples decreased. **c** The oxidation models can be selected by maturity parameters. Source data are provided as a Source Data file.

SCM, and S2, S3, and S4 ($O_2/CH_4 \geq 0.6$) belong to the IOM, which corresponds to the concentration distribution of soot shown in Fig. 1c. It was found that the selection of the oxidation model depends on the main particle type in the soot sample. For more macroscopic conditions in the flame, the nanostructures change differently. The effect of various macroscopic conditions on application of our oxidation models can be ultimately evaluated by the maturity parameter of soot samples.

Two approaches are proposed to obtain the maturity parameter for real combustion modeling. (1) In the situation that soot samples can be obtained, the maturity parameter is directly calculated from the nanostructure as proposed above. (2) In the situation that soot samples cannot be obtained, the maturity parameter can be theoretically calculated using the number density function by quantifying the carbon atom number and C/H ratio[62] of the as-formed soot in combustion modeling. The oxidation models can be selected on that basis to calculate the reaction rate in population balance equations which has been applied in counter flow flame, diffusion flame, and premixed flame[63–66]. As soot maturation progresses, the oxidation model can be changed with maturity parameters. Compared to the current modeling using empirical equations, the oxidation models provide more detail about oxidation process thereby more accurate results. This means that the soot oxidation behavior can be theoretically predicted by given the macroscopic parameters. More details can be referred to the Supplementary Discussion.

This study established three soot oxidation models applicable to different types of soot basing on the particle oxidation behavior. The corresponding oxidation behavior can be predicted by soot maturity under specific combustion conditions. The models can be used for kinetic calculations and optimize soot oxidation process in engine system and industrial furnace.

## Methods
### Soot preparation
NCPOX is a mature technology that involves a typical turbulent fuel-rich flame resulting from the combustion of gas fuels. The soot produced during the NCPOX process has properties similar to those of other soot obtained from gas fuels. The specific differences in properties have been discussed in our previous work[3,27]. The soot samples in this study were produced by a lab-scale device via the NCPOX process. The conditions used for soot production are listed in Table 1. We quantified all the reactants and products. The properties of soot were characterized. These samples have certain regularity in properties and are more suitable for comparative study than ordinary commercial carbon black.

$CH_4$ and $O_2$ reacted in a corundum tube at atmospheric pressure. The molar ratio of $H_2$ to CO at the outlet was ~2. The reaction temperature was controlled to be constant at 1200 °C. Soot was collected at the outlet through a hopper with a filter screen. The hopper was covered by alumina bulk fiber, which prevented the soot precursor from condensing in the hopper. As a result, no soot was adsorbed on the surface of the soot particles, which was verified by the Soxhlet extraction method. The samples were dried at 105 °C for 24 hours for further oxidation and characterization.

### In situ oxidation
The oxidation behaviors of individual soot particles were determined by in situ TEM. In situ aberration-corrected STEM experiments were performed using a Climate S3 in situ TEM holder (DENS Solutions Company), allowing dynamic observation at high temperatures. The oxidation reaction was confined to a tiny gas cell within 5 μm in thickness. The reaction gas cell consisted of a pair of microelectromechanical system (MEMS) chips with an amorphous silicon nitride ($SiN_x$) membrane for TEM observation in real time. A total of 0.5 mg of each soot sample was sonicated with 20 mL of ethyl alcohol for 10 min at 45°C and finally deposited on chips. It was verified that this method can yield the best concentration in the windows of the membrane. The reaction pressure was 100 mbar. The pressure was selected to control the oxidation processes at an appropriate reaction rate, facilitating the clear observation of particle evolution. The oxidation agent (0.5% $O_2$ and 99.5% $N_2$) was continuously pumped into the gas cell from the inlet. The oxidation agent was in full contact with the soot particles before the reaction. During the reaction, a very small quantity of the $CO_2$ product was pumped out. At this microscale, the diffusion effect was negligible, and the reaction was considered an intrinsic reaction.

The soot samples were heated to 400 °C from ambient temperature through chips in the vacuum. The observation position and magnification of the soot samples were adjusted. Then, the oxidation agent was pumped in until the pressure stabilized. The soot samples could not be oxidized at 400 °C. Oxidation was initiated by rapidly increasing the temperature to 900 °C at a heating rate of 10 °C s⁻¹. Images of the samples during oxidation were recorded by TEM, clearly capturing the evolution of the internal structure of the particles. The experimental process was shown in Supplementary Fig. S2. The magnifications of the images obtained during in situ oxidation were controlled below 350 kx with less electron beam irradiation, ensuring a low level of ionization of $O_2$. The continuous flow from the inlet to the outlet further minimized the effect of ionized oxygen on the reaction.

**Table 1 | Producing conditions of soot samples**

| Sample | Temperature | Pressure | Flow rate of methane | Flow rate of oxygen |
|--------|-------------|----------|---------------------|---------------------|
| S1 | 1200 °C | 1 atm | 1 NL min$^{-1}$ | 0.5 NL min$^{-1}$ |
| S2 | | | | 0.6 NL min$^{-1}$ |
| S3 | | | | 0.7 NL min$^{-1}$ |
| S4 | | | | 0.8 NL min$^{-1}$ |

### Ex situ oxidation

The oxidation of soot groups at 900 °C was analyzed by means of a thermogravimetric analyser (NETZSCH STA 2500 Regulus). Approximately 3 mg of soot sample was heated to 900 °C at a heating rate of 20 °C min$^{-1}$ in a $N_2$ atmosphere. Then, it was switched to an oxidation atmosphere. The oxidation atmosphere consisted of 0.5% $O_2$ and 99.5% $N_2$, and the concentration of $O_2$ was the same as that in the in situ oxidation processes. This lower $O_2$ concentration can prevent soot from combusting too rapidly at high temperatures. A total flow rate of 200 mL min$^{-1}$ of oxidation agent was employed to eliminate the diffusion effects. The conversion ratio, $x$, of the soot oxidation process was defined as shown in Eq. (32).

$$x = \frac{m_0 - m}{m_0 - m_\infty} \qquad (32)$$

where $m_0$ is the initial mass of the sample, $m$ is the mass of the sample at a certain moment, and $m_\infty$ is the remaining mass of the sample. The oxidation reactivity was evaluated by the reactivity index $R_{0.9}$, which was defined as follows in Eq. (33).

$$R_{0.9} = \frac{0.9}{t_{x=0.9}} \qquad (33)$$

where $t_{x=0.9}$ is the oxidation time required for $x = 0.9$.

### Image processing

TEM images were obtained for primary particle size analysis. The boundary of each primary particle was fitted to an ellipse by a manual approach. The maximum Feret's diameter, $d_{max}$, and the minimum Feret's diameter, $d_{min}$, of the ellipse were determined by means of ImageJ software. The primary particle diameter, $D_P$, was determined from the arithmetic average of $d_{max}$ and $d_{min}$. The primary particle size, $\bar{D}_P$, of the soot sample was the average of $D_P$. The specific formula is shown in the Supplementary Methods. More than 2000 particles from 4 samples were measured in this study.

The HRTEM images of the static nanostructure of soot were obtained on a ThermoFisher Talos F200X. The original images were processed by contrast enhancement, top-hat transformation, binarization, and skeletonization, and then the meaningless fringes that were shorter than the size of the aromatic cycle (0.246 nm) were removed[67]. The final image was used for fringe analysis. The example of image processing is shown in Supplementary Fig. S3. Statistical results were obtained by the same processing of multiple groups of images.

### Characterization

The C/H ratio of the soot samples was determined by an elemental analyser (VARIO EL CUBE) on a dry and ash-free basis.

The carbon structures were characterized by Raman spectroscopy (ThermoFisher DXR), which was carried out by using a He-Ne laser (0.5 mW, 455 nm). The Raman results in the first-order Raman spectrum region were deconvoluted by means of the five-band method[68] with the following bands: G (1580 cm$^{-1}$), D1 (1350 cm$^{-1}$), D2 (1610 cm$^{-1}$), D3 (1550 cm$^{-1}$), and D4 (1180 cm$^{-1}$). The distributions of different carbon structures of the soot samples are shown in the Supplementary Fig. S4. The integral band area ratio of the D band to G band ($I_{D1}/I_G$) was used to quantify the degree of graphitization of soot samples[69].

STEM characterization was performed by means of a Thermo-Fisher Themis Z microscope equipped with two aberration correctors under 300 kV. HAADF-STEM images were recorded using a convergence semi-angle of 11 mrad and inner and outer collection angles of 59 and 200 mrad, respectively. EDS was carried out using 4 in-column Super-X detectors.

### Reporting summary

Further information on research design is available in the Nature Portfolio Reporting Summary linked to this article.

## Data availability

All data that support the plots within this manuscript are available. Source Data file has also been deposited in Figshare under accession link https://doi.org/10.6084/m9.figshare.24039180[70]. Source data are provided with this paper.

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

## Acknowledgements

This work was financed by the National Natural Science Foundation of China (22178114) (F.C.W.) and the China Scholarship Council (M.G.), the Science and Technology Commission of Shanghai Municipality (22ZR1415700) (S.D.), Shanghai Rising-star Program (20QA1402400) (S.D.). Thanks for the support of the Frontiers Science Center for Materiobiology and Dynamic Chemistry and the Feringa Nobel Prize Scientist Joint Research Center at East China University of Science and Technology.

## Author contributions

M.G. and Y.J.J. conducted the experiments; M.G. designed the research, analyzed the data, and wrote the paper; L.D., Y.F.G. and W.M.Y. contributed to writing and editing; Z.H.D., G.S.Y. and F.C.W. supervised the work; F.C.W. and S.D. conceptualized and directed the project.

## Competing interests

The authors declare no competing interests.
