## [Peer Review File · Nature Communications]

Mechanism of the noncatalytic oxidation of soot using in situ transmission electron microscopyEditorial Note: Parts of this Peer Review File have been redacted as indicated to remove third-party material where no permission to publish could be obtained.

REVIEWER COMMENTS

Reviewer #1 (Remarks to the Author):

The article is extremely interesting but I have a major concern. I doubt the novelty of the manuscript would fit the high standard of the journal. However, I am sure it would fit other journals with more specific topic.

There are no major concerns on the scientific results.

Some minor comments:

- Please review the English grammar that has some mistakes and in some cases the sentences are long and hard to read;
- For the above-mentioned reason, the overall length of the manuscript is high. I would suggest to remove some of the results and place them in a Supplementary Information section.

Reviewer #2 (Remarks to the Author):

In the article by Gao et al on oxidation mechanisms of soot extracted from high temperature industrial processes, the authors bring to bear a host of, in some cases, state-of the art instrumentation to characterize in-situ and ex-situ oxidation of samples of soot of different age. The article is solid and would deserve publication in the specialized literature (e.g., Carbon, Fuel and Combustion and Flame) after substantial editing of the English that in its current form leaves a lot to be desired. However, it lacks the "earth-shaking" prerequisites of originality to deserve publication in Nature.

The diagnostics are good and comprehensive but by no means new, even in the case of ETEM; the analysis is simple and the adaptation of the models to the types of soot is empirical as is the collection of soot samples. The authors classify and detail three types of soot, all mature, but to a different extent, as revealed by the primary particle that is never smaller than 80 nm. There is no incipient soot or nucleation soot whose particle sizes are in the range of few nanometer, contrary to the authors' suggested nomenclature. The main drawback of the paper is its empirical nature: aside from some generic statements on the age of soot there is no clear recipe to correlate the relevance of the various "soot models" to the samples, let alone the combustion environment/conditions at which the soot was extracted. In the absence of such a fundamental correlation, I am left wondering when to apply any of their models to soot oxidation.

Comments of Reviewer #1

Thank you so much for your valuable comments and suggestions on our paper. We have revised the manuscript and carefully checked the grammar accordingly, and all the modifications are highlighted in yellow in the revised manuscript. The point-by-point answers to your comments and suggestions are listed below for your kind perusal.

1. The article is extremely interesting but I have a major concern. I doubt the novelty of the manuscript would fit the high standard of the journal. However, I am sure it would fit other journals with more specific topic.

Response:

Thank you so much for your appreciation of this work and your invaluable comment for improving the quality of the paper. According to current studies, we did our best to further define the research impact and refine the points of innovation. The novelty of the manuscript is discussed according to the following three factors. The novelty is also described in the revised manuscript.

(1) This study for the first time developed good correlations between quantitative mathematical models and advanced *in situ* TEM observations for the characterizations of soot oxidation, while prior *in situ* TEM studies only focused on qualitative analysis.

In situ TEM is a cutting-edge technology for high spatial resolution measurements under reaction conditions. *In situ* oxidation of soot can be divided into noncatalytic oxidation [1-5] and catalytic oxidation [6-11]. The noncatalytic oxidation mainly focused on the particle behaviours during oxidation. Table 1 lists the studies conducted

on soot noncatalytic oxidation through *in situ* TEM.

Sediako et al. [1] observed the noncatalytic oxidation behaviour of soot aggregates by ETEM for the first time. Through observations on the soot particles, it was hypothesized that soot has different oxidation modes. However, kinetic studies on these oxidation modes were not conducted. Following their studies, Sediako et al. [2] and Toth et al. [3] conducted *in situ* observations under various pressures and temperatures. Naseri et al. [4] and Dadsetan et al. [5] further discovered that the electron beams of TEM also have impacts on carbon black particles conversions under *in situ* oxidation conditions. From these limited investigations, one can find that only qualitative studies have been conducted on the soot oxidation under *in situ* conditions. Given the complexity of soot particle structures and the oxidation mechanisms, a quantitative model obtained at *in situ* conditions is highly desirable.

In the current manuscript, we conducted systematic and comparative studies by characterizing a series of soot samples. Through qualitative analysis, we found that these samples were all composed of soot particles with three distinguish type of nanostructures. Quantitative analysis was further conducted by establishing a maturity parameter to define the types of nanostructures. This enables the characterization of soot nanoparticles through quantitative analysis for the first time, while prior studies could only obtain a quasi-quantitative maturity by conducting physiochemical analysis such as Raman on soot particles under a macro scale. This largely increased the spatial resolution in terms of soot particle characterization. The oxidation mechanisms were found highly dependent on the nanostructures of particles. The maturity parameter

could quantitatively categorize the types of oxidation mechanisms based on different particle nanostructures.

Besides quantitative models for soot nanoparticles, we have for the first time proposed a mathematical approach to facilitate the kinetic calculations of soot aggregates oxidations. It was noted that prior studies were not focused on that, partly due to the complexity of soot aggregates caused by high fractal dimensions.

As summarized, this study provides quantitative and systematic analysis of the noncatalytic oxidation mode of soot particles in nanoscale, which has exceeded the depth of prior studies.

Table 1 *In situ* TEM studies on soot noncatalytic oxidation

No.	Researchers	Year	Oxidation Temp.	Pressure	Research content	Carbon Materials
1	Sediako et al. [1]	2017	550°C	1Pa	Soot aggregate oxidation behaviour using ETEM	1-decene and ethylene flames soot
2	Sediako et al. [2]	2019	1100K	1Pa	Oxidation of soot produced at different pressure	Ethylene-fueled diffusion flame soot
3	Toth et al. [3]	2019	600°C 900°C	1Pa 10Pa	The structure evolution and particle fragmentation during soot oxidation	Premixed laminar ethylene-air flat-flame soot; N990; GCB
4	Nasari et al. [4]	2020	800°C	1Pa	The difference between O ₂ and O radical oxidation of carbon black	N990
5	Dadsetan et al. [5]	2022	800°C	1Pa	The impact of particle size and electron beam to carbon black oxidation	N330; N772; N990

(2) This study for the first time observed and proved the “core-shell separation” phenomena of soot under *in situ* conditions, which developed a new cognition of the oxidation mode of soot.

The hollow structure formed by soot oxidation has been widely reported in prior studies [12]. Based on *ex situ* observations and other indirect characterizations, most

researchers believe that the hollow structure formed from the centre of the particles. However, this theory was not suitable for all situations. The understanding of the formation of hollow structure is still incomplete, because theories all lacked direct evidence obtained under *in situ* conditions.

In this study, we have discovered a new formation mechanism for the hollow structure confirmatively through reaction conditions under *in situ* TEM, which has not been reported before. The new mechanism starts with oxidation from the shell, then the core-shell separation takes place. This is different from the previously reported models. The “core-shell separation model (CSM)” obtained under *in situ* conditions in this study provides researchers with a more comprehensive understanding of soot oxidation processes. The reason for the initiation of the CSM was ascribed to the uneven distribution of oxygen-containing functional groups and PAH fragments in soot particles as revealed by energy dispersive X-ray spectroscopy. We further found that through this novel CSM oxidation process, soot particles with different structures can be synthesized. For example, a cage-structure carbon material could be formed through CSM, which exhibited strong oxidation resistance properties according to kinetic analysis as discussed before. This material can be potentially used as a storage and transportation medium for proteins according to literatures [13].

2. Please review the English grammar that has some mistakes and in some cases the sentences are long and hard to read;

Response:

Thanks a lot for your kind suggestions. We have polished the English expressions

throughout the text. We also utilized the language editing service provided by Springer Nature to satisfy the high level of Nature publication standards. The certificate is attached. The grammar has been checked, and all the improper points have been corrected. All modifications are highlighted in the revised manuscript.

[REDACTED]

Fig. 1 Editing Certificate provided by Springer Nature

3. For the above-mentioned reason, the overall length of the manuscript is high. I would suggest to remove some of the results and place them in a Supplementary Information section.

Response:

Thanks a lot for your kind suggestions. In the revised manuscript, we condensed the discussion section and highlighted the key conclusions. Some repetitive expressions

were removed. Detailed descriptions of the nanostructures in section 3.1 were deleted. Some detailed information of in situ TEM experiment, the Raman fitting processes, and the distributions of different carbon structures of soot samples are shown in the Supplementary Information. The discussion of the *ex situ* oxidation results was refined. The methods section has been modified to highlight the novelty.

Comments of Reviewer #2

Thank you so much for your invaluable comments and suggestions, which are extremely helpful for us to significantly improve the paper and define the important guiding significance of our research. We have revised the manuscript according to your comments. We conducted a detailed analysis and established new parameters to quantify the maturity of our samples. Our point-by-point answers are listed below for your kind perusal.

1. "... after substantial editing of the English that in its current form leaves a lot to be desired."

Response:

Thank you so much for your kind suggestion. We have polished the English expressions throughout the text. We have tried our best to improve the language. We also utilized the language editing service provided by Springer Nature to satisfy the high level of Nature publication standards. The certificate is attached. The grammar has been checked, and all the improper points have been corrected. All modifications are highlighted in the revised manuscript.

[REDACTED]

Fig. 1 Editing Certificate provided by Springer Nature

2. However, it lacks the “earth-shaking” prerequisites of originality to deserve publication in Nature.

Response:

Thanks a lot for your invaluable comment. According to current studies, we did our best to further define the research impact and refine the points of innovation. The novelty of the manuscript is discussed according to the following three factors. The novelty is also highlighted in the revised manuscript.

(1) This study for the first time developed good correlations between quantitative mathematical models and advanced *in situ* TEM observations for the characterizations of soot oxidation, while prior *in situ* TEM studies only focused

on qualitative analysis.

In situ TEM is a cutting-edge technology for high spatial resolution measurements under reaction conditions. *In situ* oxidation of soot can be divided into noncatalytic oxidation [1-5] and catalytic oxidation [6-11]. The noncatalytic oxidation mainly focused on the particle behaviours during oxidation. Table 1 lists the studies conducted on soot noncatalytic oxidation through *in situ* TEM.

Sediako et al. [1] observed the noncatalytic oxidation behaviour of soot aggregates by ETEM for the first time. Through observations on the soot particles, it was hypothesized that soot has different oxidation modes. However, kinetic studies on these oxidation modes were not conducted. Following their studies, Sediako et al. [2] and Toth et al. [3] conducted *in situ* observations under various pressures and temperatures. Naseri et al. [4] and Dadsetan et al. [5] further discovered that the electron beams of TEM also have impacts on carbon black particles conversions under *in situ* oxidation conditions. From these limited investigations, one can find that only qualitative studies have been conducted on the soot oxidation under *in situ* conditions. Given the complexity of soot particle structures and the oxidation mechanisms, a quantitative model obtained at *in situ* conditions is highly desirable.

In the current manuscript, we conducted systematic and comparative studies by characterizing a series of soot samples. Through qualitative analysis, we found that these samples were all composed of soot particles with three distinguish type of nanostructures. Quantitative analysis was further conducted by establishing a maturity parameter to define the types of nanostructures. This enables the characterization of

soot nanoparticles through quantitative analysis for the first time, while prior studies could only obtain a quasi-quantitative maturity by conducting physiochemical analysis such as Raman on soot particles under a macro scale. This largely increased the spatial resolution in terms of soot particle characterization. The oxidation mechanisms were found highly dependent on the nanostructures of particles. The maturity parameter could quantitatively categorize the types of oxidation mechanisms based on different particle nanostructures.

Besides quantitative models for soot nanoparticles, we have for the first time proposed a mathematical approach to facilitate the kinetic calculations of soot aggregates oxidations. It was noted that prior studies were not focused on that, partly due to the complexity of soot aggregates caused by high fractal dimensions.

As summarized, this study provides quantitative and systematic analysis of the noncatalytic oxidation mode of soot particles in nanoscale, which has exceeded the depth of prior studies.

Table 1 *In situ* TEM studies on soot noncatalytic oxidation

No.	Researchers	Year	Oxidation Temp.	Pressure	Research content	Carbon Materials
1	Sediako et al. [1]	2017	550°C	1Pa	Soot aggregate oxidation behaviour using ETEM	1-decene and ethylene flames soot
2	Sediako et al. [2]	2019	1100K	1Pa	Oxidation of soot produced at different pressure	Ethylene-fueled diffusion flame soot
3	Toth et al. [3]	2019	600°C 900°C	1Pa 10Pa	The structure evolution and particle fragmentation during soot oxidation	Premixed laminar ethylene-air flat-flame soot; N990; GCB
4	Naseri et al. [4]	2020	800°C	1Pa	The difference between O ₂ and O radical oxidation of carbon black	N990
5	Dadsetan et al. [5]	2022	800°C	1Pa	The impact of particle size and electron beam to	N330; N772; N990

(2) This study for the first time observed and proved the “core-shell separation” phenomena of soot under *in situ* conditions, which developed a new cognition of the oxidation mode of soot.

The hollow structure formed by soot oxidation has been widely reported in prior studies [12]. Based on *ex situ* observations and other indirect characterizations, most researchers believe that the hollow structure formed from the centre of the particles. However, this theory was not suitable for all situations. The understanding of the formation of hollow structure is still incomplete, because theories all lacked direct evidence obtained under *in situ* conditions.

In this study, we have discovered a new formation mechanism for the hollow structure confirmatively through reaction conditions under *in situ* TEM, which has not been reported before. The new mechanism starts with oxidation from the shell, then the core-shell separation takes place. This is different from the previously reported models. The “core-shell separation model (CSM)” obtained under *in situ* conditions in this study provides researchers with a more comprehensive understanding of soot oxidation processes. The reason for the initiation of the CSM was ascribed to the uneven distribution of oxygen-containing functional groups and PAH fragments in soot particles as revealed by energy dispersive X-ray spectroscopy. We further found that through this novel CSM oxidation process, soot particles with different structures can be synthesized. For example, a cage-structure carbon material could be formed through CSM, which exhibited strong oxidation resistance properties according to kinetic

analysis as discussed before. This material can be potentially used as a storage and transportation medium for proteins according to literatures [13].

3. The diagnostics are good and comprehensive but by no means new, even in the case of ETEM;

Response:

Thanks a lot for your kind comment. As discussed above, we would like to highlight that although *in situ* TEM has been conducted on the noncatalytic oxidation of soot, they are mainly focusing on the qualitative analysis. As compared, this study for the first time developed good correlations between quantitative mathematical models and advanced *in situ* TEM observations for the characterizations of soot oxidation, and largely increased the depth and generality of the study.

To elaborate, we have established a quantitatively defined maturity parameter to categorize the nanostructure of soot particles at various maturity levels, which in turn leads to different oxidation mechanisms. We have also established simplified models for soot aggregates, which facilitated the calculation of reaction kinetics for soot oxidation under *in situ* TEM observations. These innovations have greatly improved the utilization efficiency of *in situ* TEM and maximized the technical advantages of *in situ* TEM.

4. the analysis is simple and the adaptation of the models to the types of soot is empirical as is the collection of soot samples.

Response:

Thanks a lot for your invaluable comment. It is very helpful, and we have further

improved our work according to your suggestions. We realized that the analysis of the properties of soot particles was not detailed enough. Therefore, we have established a new parameter to quantify the maturity of soot and related the properties to general soot particles. The sample preparation process was also described in detail for researchers to repeat in the supplemental file.

The followings are detailed discussions, we have added these into the revised manuscript:

(1) “In previous studies, the particle was treated as spheroidal in most cases of *in situ* oxidation of carbonaceous particles. However, the morphology of soot aggregates is too complex. There are 4 forms of soot aggregates, which are spheroidal, ellipsoidal, linear, and branched, as shown in Fig. 2 [24]. Most industrial soot is in the branched form. This results in soot aggregates that cannot be treated as spheroidal. Therefore, the existing kinetic model of carbonaceous particles was unsuitable for soot aggregates. More applicable models for soot aggregate kinetics are needed.

[REDACTED]

Fig. 2 Morphology of soot aggregates [24]

[REDACTED]

Fig. 3 Detailed particle model by Lindberg et al. [25]

A detailed particle model was published by Lindberg et al. [25] in 2019. However, the model was not developed for oxidation processes. We have tried to develop the model to be suitable for the oxidation of soot aggregates. The parameters of the model are described in Fig. 3. The particle centre-to-neck distance, x_{ij} , and primary volume, v_i , can be described by the following equations.

$$x_{ij} = \frac{d_{ij}^2 - r_j^2 + r_i^2}{2d_{ij}} \quad \text{Eq. (8)}$$

$$v_i = \frac{4}{3}\pi r_i^3 - \frac{1}{3}\pi \sum_j (2r_i^3 + x_{ij}^3 - 3r_i^2 x_{ij}) \quad \text{Eq. (9)}$$

$$x = \frac{v_0 - v_i}{v_0 - v_\infty} \quad \text{Eq. (10)}$$

Parameters such as the equivalent radius and centre coordinates can be determined through frame-by-frame processing of experimental images. However, the expression of the conversion ratio is too complicated when it is further used for oxidation kinetics calculations. Eq. (8) and Eq. (9) should be applied to Eq. (10), which makes it too difficult to integrate.

As Octave Levenspiel described in *Chemical Reaction Engineering*, we are seeking a good engineering model that is close to reality but without excessive mathematical complexity. For this, we propose a simplified approach to solve the

problem, as shown in Fig. 4. After simplification, the parameters of the primary particle are no longer used, and only the parameters of the equivalent sphere are used in the calculation.

Fig. 4 Simplification process of soot aggregates

$$x = 1 - \frac{\sum s_i^{\frac{3}{2}}}{\sum s_{0i}^{\frac{3}{2}}} \quad \text{Eq. (11)}$$

$$x = 1 - \frac{S^{\frac{3}{2}}}{S_0^{\frac{3}{2}}} \quad \text{Eq. (12)}$$

As shown in Eq. (11), the simplified expression of soot aggregates has been greatly improved from the detailed particle model. Notably, the conversion expression of aggregates is similar to that of single particles (Eq. (12)). This means that the models of the aggregates can be calculated by using the same kinetics method used for single particles. This is the main advantage of the models. This advantage is more evident in complicated oxidation models, such as the CSM:

$$x = \frac{\sum \left(s_0^{\frac{3}{2}} - s^{\frac{3}{2}} \right)}{\sum \left(s_0^{\frac{3}{2}} + V'' \right)} \quad \text{Eq. (13)}$$

$$x = 1 - \frac{V}{V_0} = \frac{S_0^{\frac{3}{2}} - S^{\frac{3}{2}}}{S_0^{\frac{3}{2}} + V''} \quad \text{Eq. (14)}$$

These mathematical expressions originated from the modelling of the experimental results. The semiquantitative reaction rates were calculated based on various soot aggregate models. The reaction rates obtained were in good agreement with the experimental results. In addition, it is more intuitive to describe the evolution of particles than the results of *in situ* oxidation by TEM. The establishment of a simplified expression makes it possible for oxidation models to be applied to practical engineering problems. This simplified approach facilitated rapid and precise calculation of the reaction rates for various types of soot by the researchers.”

(2) “In current studies, the maturity level is mainly used to describe the relative average properties of a series of samples [15, 26-29]. However, the oxidation modes depend on the individual particle nanostructures. We established a parameter to quantify the maturity level of different individual particles in the revised manuscript.

Soot maturity mainly depends on two independent parameters, the C/H ratio and the primary particle size [14]. The C/H ratio reflects the degree of dehydrogenation and carbonization at high temperature. The particle size reflects the degree of surface growth. However, determining the maturity by particle size alone is not always possible. In fact, the particle size reflects the effect of growth on maturity. Therefore, we considered quantifying the maturity in terms of the C/H ratio with size correction.

As particles mature, the arrangement of the PAHs changes, and they gradually grow to form a graphitic shell [15]. Alfè et al. [16, 17] reported that the fringe length increased from incipient soot to mature soot. Therefore, the maturity process can be predicted by the fringe length. The carbon atom number of PAH molecules can be

calculated by Eq. (1) [18].

$$C_{\text{PAH},i} \approx \left[(L_{a,i} + 0.211) / 0.193 \right]^2 \quad \text{Eq. (1)}$$

where $L_{a,i}$ is the fringe length of the i th PAH determined by HRTEM. The carbon-to-

hydrogen ratio of $\left(\frac{C}{H} \right)_{\text{PAH},i}$ can be expressed as Eq. (2) [19].

$$\left(\frac{C}{H} \right)_{\text{PAH},i} = \left(\frac{C_{\text{PAH},i}}{6} \right)^{0.43} \quad \text{Eq. (2)}$$

We found that Eq. (2) was suitable for plane PAHs. Therefore, it has a good prediction ability for immature soot. Upon maturation, defects can cause PAHs to curve from a two-dimensional structure to a three-dimensional structure. The mature soot contains more curved PAHs, which have higher C/H ratios. Here, we established a dimensionless parameter, τ , to correct the hydrogen number of curved PAHs. As shown in Fig. 5 (a)(b), the H atoms were only distributed on the edge of the PAH. The actual number of H atoms was related to the length of the sides. In HRTEM, τ reflects the tortuosity of a fringe. It can be defined as the ratio of the fringe length to the distance between the two endpoints, as shown in Fig. 5 (c) and Eq. (3).

$$\tau = \frac{L_a}{D} \quad \text{Eq. (3)}$$

Fig. 5 The structures of (a) plane PAH and (b) curved PAH and (c) the curved fringe in the particle

The corrected number of hydrogen atoms, $H_{\text{PAH},i}$, can be expressed as Eq. (4).

$$H_{\text{PAH},i} = \frac{C_{\text{PAH},i}}{\tau \left(\frac{C}{H} \right)_{\text{PAH},i}} \quad \text{Eq. (4)}$$

For a PAH with fewer than 6 rings, the molecule is too small to bend. Curved PAHs often form due to the five-membered carbon rings inside them. We selected corannulene ($\text{C}_{20}\text{H}_{10}$) as the first molecule to be curved.

Fig. 6 The molecular structure of corannulene

As shown in Fig. 6, corannulene is a PAH with nearly circular symmetry and 20 carbon atoms. When the carbon atom number of partially matured soot and mature soot

was below 20, $\tau = 1$. Therefore, we defined the maturity $M_{\text{particle},j}$ of a single particle j as the ratio of the number of C atoms to the corrected number of H atoms of all PAHs, as shown in Eq. (5).

$$M_{\text{particle},j} = \frac{\sum_i C_{\text{PAH},i}}{\sum_i H_{\text{PAH},i}} = \frac{\tau \sum_i (L_{a,i} + 0.211)^2}{2.404 \sum_i (L_{a,i} + 0.211)^{1.14}} \quad \text{Eq. (5)}$$

When $L_{a,i} < 0.652$, $\tau = 1$; $L_{a,i} \geq 0.652$, $\tau = \frac{L_a}{D}$. The same method can be used to

determine the average maturity of multiple particles, as shown in Eq. (6).

$$M_{\text{sample}} = \frac{\tau \sum_{i,j} (L_{a,i,j} + 0.211)^2}{2.404 \sum_{i,j} (L_{a,i,j} + 0.211)^{1.14}} \quad \text{Eq. (6)}$$

When $L_{a,i,j} < 0.652$, $\tau = 1$; $L_{a,i,j} \geq 0.652$, $\tau = \frac{L_a}{D}$. To determine the exact value of

M_{sample} , we performed these calculations for more particles. We analysed the M_{sample} of 4 samples in this study and compared the results of elemental analysis. The relationship between the two is shown in Fig. 7. Since the curve passes through the origin, the allometric function was performed.

Fig. 7 The fitting line of the C/H ratio and M and the range of soot at different maturity levels

$$\left(\frac{C}{H}\right)_{\text{soot}} = 1.72M_{\text{sample}}^{1.07} \quad \text{Eq. (7)}$$

Due to the different analytical principles of instruments, the macroscopic C/H ratio obtained through Eq. (7) may be underestimated. The main significance of M_{sample} lies in reflecting the extent of growth and carbonization of the sample. It is generally recognized that the C/H ratio of incipient soot is 1.4~2.5 and that of mature soot is 8~20 [20-23]. By using Eq. (7), we can calculate the M_{sample} value of incipient soot to be 0.83~1.42 and that of mature soot to be 4.21~9.91. We can correlate the 3 types of soot with the general sample by calculating the M_{sample} value. The young soot, which was named incipient soot in the last manuscript, was in the early stage of carbonization. In this stage, the primary focus is on the growth of soot particles. Soot transitions from coalescence to agglomeration and forms a particle structure from liquid-like incipient soot. The young soot corresponds to the soot samples with a C/H ratio of 2.5~4.1. The partially matured soot is in the middle of carbonization. This is a stage that mainly

involves dehydrogenation and surface reaction. The particles begin to form graphitic shells. The arrangement of PAH crystals changes. The partially matured soot corresponds to the soot samples with a C/H ratio of 4.1~8. The mature soot is in the late stages of carbonization. The particles have formed smooth graphitized shells with a highly spherical shape. Mature soot forms larger aggregates in this stage. The mature soot corresponds to the soot samples with a C/H ratio of 8~20. The calculated maturity stage of the sample is consistent with the corresponding nanostructure determined by TEM, which further proves the rationality of using M_{sample} .

M_{sample} relates the sample of this study to soot in general. Because the maturation process of soot is always the same, the maturity stage of any sample and the corresponding oxidation model can be determined by means of M_{sample} .”

(3) “The noncatalytic partial oxidation process (NCPOX) is a mature technology that has been industrialized after more than ten years of research. It involves a typical high-temperature enriched flame. The soot samples were produced by a lab-scale NCPOX device. All the products in the experiment were quantified, and their properties were analysed. The soot samples collected from solid-phase products under different working conditions are listed in Table 2. This approach can serve as a valuable reference for researchers. These samples have certain regularity in their properties and are more suitable for comparative study than ordinary commercial carbon black. By applying the maturity parameter proposed above in combination with the TEM, Raman, and TGA results, the samples were well correlated with the general samples.”

Table 2 Producing conditions of soot samples

Sample	Temperature	Pressure	Residence time	Flow rate of methane	Flow rate of oxygen
S1	1200°C	1 atm	2.01s	1 NL/min	0.5 NL/min
S2			1.58s		0.6 NL/min
S3			1.36s		0.7 NL/min
S4			1.23s		0.8 NL/min

5. The authors classify and detail three types of soot, all mature, but to a different extent, as revealed by the primary particle that is never smaller than 80 nm. There is no incipient soot or nucleation soot whose particle sizes are in the range of few nanometer, contrary to the authors' suggested nomenclature.

Response:

Thank you so much for your invaluable comment. We further analysed the exact maturity stage of different types of samples, and the three types of particles were renamed as young soot, partially matured soot, and mature soot.

The incipient soot with a size of a few nanometres (less than 10 nm) was almost transparent to the electron beam [30]. This was due to the incipient soot showing obvious liquid-like properties, as shown in Fig. 8 [31]. These particles had not yet formed a complete solid particle structure, and the shape and internal nanostructure were irregular. Therefore, the effect of structural differences on oxidation patterns could not be determined. In addition, incipient soot exhibits active oxidation reactivity in a high-temperature furnace and is easier to eliminate. In this study, the solid products and liquid products were separated at higher temperatures. No soot precursors or liquid-like incipient soot (<10 nm) were observed in the TEM images. The particle size observed ranges from ~26 nm to ~391 nm.

[REDACTED]

Fig. 8 Liquid-like incipient soot particles [30].

Therefore, this study focused on soot particles that formed solid particle structures. The oxidation reactivities of these soot samples were much lower and more worthy of study. According to M_{sample} , the maturity of the samples in this study was higher than that of incipient soot. It is necessary to rename the three types of soot. The terminology describing soot at different stages in the fields of combustion, industrial chemicals and atmospheric and environmental science [32] is shown in Table 3.

Table 3 Nomenclature of soot at different maturity levels

Stage 1	Stage 2	Stage 3
Nascent soot	Young soot	
Incipient soot	Partially aged soot	
	Partially carbonized soot	Mature soot
	Partially graphitized soot	
	Partially matured soot	

According to the maturity parameters proposed above and the nanostructures of particles, the three types of soot were renamed young soot, partially matured soot, and mature soot. The particle size distributions and M_{sample} of the collected samples are shown in Table 4. M_{sample} can better predict the maturity of the samples compared to particle size.

Table 4 The particle size distributions of different soot types

Soot type	Main particle size range	M_{sample}
Young soot	26~100 nm	1.42~2.26
Partially matured soot	53~143 nm	2.26~4.21
Mature soot	58~281 nm	4.21~9.91

Fig. 9 Classification and physical model of soot particles

As shown in Fig. 9, young soot exhibited irregular PAH crystalline arrangements and had not yet formed graphitic shells. These soot particles were in the coalescence stage. Some small incipient soot formed larger young particles, gradually formed solid particles, and then began to agglomerate and mature. For partially matured soot, these particles formed a few concentrically oriented graphitized shells, which is a sign of partially matured soot. As maturation continued, surface growth and dehydrogenation occurred continuously, ultimately resulting in the formation of mature soot with a smooth and densely packed shell structure. M_{sample} was derived from the nanostructure of the sample. We strongly suggest using the maturity parameter proposed in this paper to quantify the maturity level of soot particles.

6. The main drawback of the paper is its empirical nature: aside from some generic statements on the age of soot there is no clear recipe to correlate the relevance of the various “soot models” to the samples, let alone the combustion environment/conditions at which the soot was extracted. In the absence of such a fundamental correlation, I am left wondering when to apply any of their models to soot oxidation.

Response:

Thank you so much for your invaluable comment. Your suggestion is very pertinent, and we greatly appreciate it. As explained above, we have placed significant emphasis on this aspect and have made substantial improvements and conducted detailed analyses. We proposed a maturity parameter, M_{sample} , derived from the nanostructure of soot particles, which can reflect the degree of growth and carbonization. We quantified the age of different particles through M_{sample} and related it to the C/H ratio of general particles. Based on M_{sample} , we determined the applicable range of different oxidation models, as listed in Table 5.

In this study, we have taken soot samples produced from various working conditions. From *in situ* TEM, we observed that these samples were all composed with soot particles categorized in three classes, namely young soot, partially matured soot and mature soot. Additionally, these different classes of soot particles can be further quantitatively categorized by a maturity parameter as M_{sample} . Considering the mechanisms for soot maturation is generally the same, this quantified maturity parameter can be extended to other soot samples prepared in different conditions. It was

noted that due to time and funding limitations, we cannot exhaust all possible soot samples to prove that trend. But based on the quantitative analysis approach described above, we believe that the current work should have good generality. Future works have been planned to largely extend the characterizations of the current soot samples and prove this trend in an even larger scale.

Table 5. Soot types and applicable range of oxidation models

Soot type	C/H	M	I_{D1}/I_G	Oxidation Model	Reaction characteristic
Young soot	2.5~4.1	1.42~2.26	~3.87	SCM	The reaction rate is fast and gradually slows down as the reaction progresses.
Partially matured soot	4.1~8	2.26~4.21	~3.56	IOM	The reaction rate is fast in the early stage and gradually slowed down in the later stage.
Mature soot	8~20	4.21~9.91	~2.87	SCM	The reaction rate is slow and gradually increases as the reaction progresses.

References

1. Sediako, A.D., et al., Real-time observation of soot aggregate oxidation in an Environmental Transmission Electron Microscope. *Proceedings of the Combustion Institute*, 2017. 36(1): p. 841-851.
2. Sediako, A.D., et al., In Situ Imaging Studies of Combustor Pressure Effects on Soot Oxidation. *Energy and Fuels*, 2019. 33(2): p. 1582-1589.
3. Toth, P., et al., Real-time, in situ, atomic scale observation of soot oxidation. *Carbon*, 2019. 145: p. 149-160.
4. Naseri, A., et al., In-situ studies of O₂ and O radical oxidation of carbon black using thermogravimetric analysis and environmental transmission electron microscopy. *Carbon*, 2020. 156: p. 299-308.
5. Dadsetan, M., A. Naseri, and M.J. Thomson, Real-time observation and quantification of carbon black oxidation in an environmental transmission electron microscope: Impact of particle size and electron beam. *Carbon*, 2022. 190: p. 1-9.
6. Simonsen, S.B., et al., Direct observations of ceo₂-catalyzed soot oxidation at the nano-scale using environmental transmission electron microscopy. *SAE International Journal of Materials and Manufacturing*, 2009. 1(1): p. 199-203.
7. Serve, A., et al., Investigations of soot combustion on yttria-stabilized zirconia by environmental transmission electron microscopy (ETEM). *Applied Catalysis A: General*, 2015. 504: p. 74-80.
8. Kamatani, K., et al., Direct observation of catalytic oxidation of particulate matter using in situ TEM. (2045-2322 (Electronic)).
9. Gardini, D., et al., Visualizing the mobility of silver during catalytic soot oxidation. *Applied Catalysis B: Environmental*, 2016. 183: p. 28-36.
10. Gao, Y., et al., Aggregation and redispersion of silver species on alumina and sulphated alumina supports for soot oxidation. *Catalysis Science & Technology*, 2017. 7(16): p. 3524-3530.
11. Aneggi, E., et al., In situ environmental HRTEM discloses low temperature carbon soot oxidation by ceria–zirconia at the nanoscale. *Chemical Communications*, 2019. 55(27): p. 3876-3878.
12. Singh, M., et al., Nanostructure changes in diesel soot during NO₂–O₂ oxidation under diesel particulate filter-like conditions toward filter regeneration. *International Journal of Engine Research*, 2019. 20(8-9): p. 953-966.
13. Amornwachirabodee, K., et al., Oxidized Carbon Black: Preparation, Characterization and Application in Antibody Delivery across Cell Membrane. (2045-2322 (Electronic)).
14. Kholghy, M.R., A. Veshkini, and M.J. Thomson, The core–shell internal nanostructure of soot – A criterion to model soot maturity. *Carbon*, 2016. 100: p. 508-536.
15. Johansson, K.O., et al., Evolution of maturity levels of the particle surface and bulk during soot growth and oxidation in a flame. *Aerosol Science and Technology*, 2017. 51(12): p. 1333-1344.
16. Alfè, M., et al., Structure–property relationship in nanostructures of young and mature soot in premixed flames. *Proceedings of the Combustion Institute*, 2009. 32(1): p. 697-704.
17. Alfè, M., et al., The effect of temperature on soot properties in premixed methane flames. *Combustion and Flame*, 2010. 157(10): p. 1959-1965.

18. Teini, P.D., D.M.A. Karwat, and A. Atreya, Observations of nascent soot: Molecular deposition and particle morphology. *Combustion and Flame*, 2011. 158(10): p. 2045-2055.
19. Commodo, M., et al., On the early stages of soot formation: Molecular structure elucidation by high-resolution atomic force microscopy. *Combustion and Flame*, 2019. 205: p. 154-164.
20. Harris, S.J. and A.M. Weiner, Chemical Kinetics of Soot Particle Growth. *Annual Review of Physical Chemistry*, 1985. 36(1): p. 31-52.
21. Ciajolo, A., et al., Spectroscopic and compositional signatures of pah-loaded mixtures in the soot inception region of a premixed ethylene flame. *Symposium (International) on Combustion*, 1998. 27(1): p. 1481-1487.
22. Li, J. and S. Yu, Soot particles analysis in laminar premixed propane/oxygen (C₃H₈/O₂) flames using published measurement data. *China Particuology*, 2003. 1(4): p. 168-171.
23. Russo, C., A. Tregrossi, and A. Ciajolo, Dehydrogenation and growth of soot in premixed flames. *Proceedings of the Combustion Institute*, 2015. 35(2): p. 1803-1809.
24. Herd, C.R., G.C. McDonald, and W.M. Hess, Morphology of Carbon-Black Aggregates: Fractal Versus Euclidean Geometry. *Rubber Chemistry and Technology*, 1992. 65(1): p. 107-129.
25. Lindberg, C.S., et al., A detailed particle model for polydisperse aggregate particles. *Journal of Computational Physics*, 2019. 397: p. 108799.
26. Michelsen, H.A., Effects of maturity and temperature on soot density and specific heat. *Proceedings of the Combustion Institute*, 2021. 38(1): p. 1197-1205.
27. Baldelli, A., et al., On determining soot maturity: A review of the role of microscopy-and spectroscopy-based techniques. *Chemosphere*, 2020. 252.
28. Leschowski, M., et al., Combination of LII and extinction measurements for determination of soot volume fraction and estimation of soot maturity in non-premixed laminar flames. *Applied Physics B*, 2015. 119: p. 685-696.
29. Yon, J., et al., Revealing soot maturity based on multi-wavelength absorption/emission measurements in laminar axisymmetric coflow ethylene diffusion flames. *Combustion and Flame*, 2021. 227: p. 147-161.
30. Abid, A.D., et al., Size distribution and morphology of nascent soot in premixed ethylene flames with and without benzene doping. *Proceedings of the Combustion Institute*, 2009. 32(1): p. 681-688.
31. Zhao, B., K. Uchikawa, and H. Wang, A comparative study of nanoparticles in premixed flames by scanning mobility particle sizer, small angle neutron scattering, and transmission electron microscopy. *Proceedings of the Combustion Institute*, 2007. 31(1): p. 851-860.
32. Michelsen, H.A., et al., A Review of Terminology Used to Describe Soot Formation and Evolution under Combustion and Pyrolytic Conditions. *ACS Nano*, 2020. 14(10): p. 12470-12490.

REVIEWERS' COMMENTS

Reviewer #1 (Remarks to the Author):

The article has been modified and highly improved. The quality is quite high but I doubt the novelty is that high for Nature Communication. I would suggest alternative journals.

Reviewer #2 (Remarks to the Author):

I have read the revised version of the article by Gao et al on oxidation mechanisms of soot extracted from high temperature industrial processes. The revision is essentially a reorganization of the paper by shifting some figures to the supplemental materials with some improvement on the English. The "rebuttal" is a lengthy restatement of what was in the original submission.

So, I am afraid I have to repeat what I already stated in the first review. I thought originally that "it lacks the prerequisites of originality to deserve publication in Nature", even though it "is solid and would deserve publication in the specialized literature (e.g., Carbon, Fuel and Combustion and Flame)". Reviewer No. 1 had the same opinion. I have not changed my opinion after the revisions.

One shortcoming that I also pointed out originally is that the authors do not even attempt to correlate soot samples and oxidation state or, as they put it, a maturity parameter with critical macroscopic parameters of a combustion environment such as residence time, temperature and oxidizer concentration. This fundamental information would make their work much more valuable especially to modelers who have no information on the soot nanostructure. Without it, they would not know when to apply their model. This is different from providing some details of their experimental conditions as in Table 2 in the rebuttal. As a modeler, I want to know what to apply to a diffusion flame or a premixed flame or whatever environment I am trying to model.

This empiricism is another limitation that makes it unsuitable for the standards of Nature, but, perhaps, I have idealized such standards.

Comments of Reviewer #1

1.The article has been modified and highly improved. The quality is quite high but I doubt the novelty is that high for Nature Communication. I would suggest alternative journals.

Response:

We appreciate your affirmation of our work. Thank you very much for your comments and suggestions.

We have further expanded the applicability of the model according to the comments of Reviewer #2.

The novelty of this paper is highlighted as follows: **(1)** This study developed good correlations between quantitative mathematical models and advanced in situ TEM observations for the characterizations of soot oxidation. **(2)** A methodological approach for determining oxidation behavior based on soot maturity was proposed, enabling the prediction of different oxidation models for various soot particles. **(3)** Besides, this study observed and proved the “core-shell separation” phenomena of soot under in situ conditions, and a new cognition of the oxidation mode of soot is disclosed in this work for the first time.

Comments of Reviewer #2

Thank you so much for your invaluable comments and suggestions. We have further discussed the applicability of the models in flame environments, to guide the modelers on how to apply the models to real combustion calculations. The expanding applicability has been added in our second revised manuscript and marked in yellow. Our point-by-point answers are listed below for your kind perusal.

1.I have read the revised version of the article by Gao et al on oxidation mechanisms of soot extracted from high temperature industrial processes. The revision is essentially a reorganization of the paper by shifting some figures to the supplemental materials with some improvement on the English. The “rebuttal” is a lengthy restatement of what was in the original submission.

So, I am afraid I have to repeat what I already stated in the first review. I thought originally that “it lacks the prerequisites of originality to deserve publication in Nature”, even though it “is solid and would deserve publication in the specialized literature (e.g., Carbon, Fuel and Combustion and Flame)”. Reviewer No. 1 had the same opinion. I have not changed my opinion after the revisions.

Response:

Thank you so much for your kind comment. Sorry for the lengthy response last time. We would like to highlight that a lot more work has been done in our first revision with more results and discussions added rather than simply a restatement of our work. To put it simply, the newer and deeper analyses in our manuscript are

highlighted as the following novelties.

The first novelty of this study is: (1) for the first time, good correlations between quantitative mathematical models and advanced in situ TEM observations were developed for the characterizations of soot oxidation. To establish this point, we have further analysed the differences between oxidation models in more detail and verified the models with experimental results. We have also condensed the scattered figures in our original submitted manuscript (Fig. 14-18 in the original submitted manuscript) into our revised Fig. 5 in the revised manuscript (shown as Fig. R1 below). As can be seen, the differences between soot particle structures and their oxidation models have been demonstrated in a much clearer way.

Fig. R1 Proposed models of different types of soot particles.

The second novelty of this study is the methodological approach for determining oxidation behavior based on soot maturity. To establish this point, we have added a new maturity parameter in the revision to quantify the nanostructure of soot. The maturity parameter reflects the structural evolution during soot aging and can be applied to a variety of particles, as shown in the newly added Fig. 2 in the revised manuscript (shown as Fig. R2 below). The maturity parameter correlates the soot property with the oxidation model, enabling the prediction of different oxidation models for various soot particles.

Fig. R2 The correction and the result of maturity parameter.

The third novelty of this study is that we observed and proved the “core-shell separation” phenomena of soot under in situ conditions, and a new cognition of the oxidation mode of soot is disclosed for the first time. We have emphasized this point in the revised manuscript.

2. One shortcoming that I also pointed out originally is that the authors do

not even attempt to correlate soot samples and oxidation state or, as they put it, a maturity parameter with critical macroscopic parameters of a combustion environment such as residence time, temperature and oxidizer concentration. This fundamental information would make their work much more valuable especially to modelers who have no information on the soot nanostructure. Without it, they would not know when to apply their model. This is different from providing some details of their experimental conditions as in Table 2 in the rebuttal. As a modeler, I want to know what to apply to a diffusion flame or a premixed flame or whatever environment I am trying to model.

This empiricism is another limitation that makes it unsuitable for the standards of Nature, but, perhaps, I have idealized such standards.

Response:

Thank you so much for your invaluable comment. We have further discussed and expanded the applicability of the oxidation models to the real combustion environment using better correlations between macroscopic parameters and nanostructures of soot. We would like to highlight that the samples characterized in this study (namely S1, S2, S3, and S4) were prepared using changing O_2/CH_4 ratio. In order to better correlate this macroscopic parameter with our model, we have replotted our Fig. 1c as shown in Fig. R3. The O_2/CH_4 ratio has a strong negative linear correlation with the fraction of mature soot, while the fractions of young soot and partially matured soot generally increase with O_2/CH_4 . More quantified relationships are shown in Fig. R4. As can be seen, increased O_2/CH_4 ratio leads to

decreased maturities. The oxidation models of samples at each O_2/CH_4 can also be distinguished in Fig. R4. It was found that the selection of the oxidation model depends on the main particle type in the soot sample. We have revised Fig. 1c in our manuscript and added the new Fig. R4 in the revised manuscript.

Fig. R3 The distribution of three soot types in samples

Fig. R4 The relationship between the O_2/CH_4 ratio and the maturity parameter

For more macroscopic conditions in the flame, the nanostructures change differently. The effect of various macroscopic conditions can be ultimately valued by the maturity parameter, thereby choosing correlated oxidation models.

In the situation that soot samples cannot be obtained, the maturity parameter can be theoretically calculated using the number density function by quantifying the

carbon atom number and C/H ratio [1] of the as-formed soot in combustion modelling. The oxidation models can be selected on that basis to calculate the reaction rate in population balance equations, which has been widely applied in counter flow flame, diffusion flame, and premixed flame [2-5]. As the soot aging proceeds, the oxidation model can be changed with maturity parameters. Compared to the previous modelling using empirical equations from the literatures, the proposed oxidation models provide more details about oxidation process thereby more accurate results. This means that the soot oxidation behavior can be theoretically predicted given the macroscopic parameters. Please refer to the **Supplementary Discussion** for details.

This study established three soot oxidation models applicable to different types of soot basing on the particle oxidation behavior. The corresponding oxidation behavior can be predicted by soot maturity under specific combustion conditions. The models can be used for kinetic calculations and optimize soot oxidation process in engine system and industrial furnace.

It should be noted that the soot formation is very complicated [6, 7] and more details of the effect of macroscopic conditions on soot nanostructure and its oxidation process can be done using our developed methods in the future.

References

1. Salenbauch, S., et al., *Detailed particle nucleation modeling in a sooting ethylene flame using a Conditional Quadrature Method of Moments (CQMOM)*. Proceedings of the Combustion Institute, 2017. **36**(1): p. 771-779.
2. Wu, S., et al., *A joint moment projection method and maximum entropy approach for simulation of soot formation and oxidation in diesel engines*. Applied Energy, 2020. **258**.
3. Salenbauch, S., et al., *Detailed modeling of soot particle formation and comparison to optical diagnostics and size distribution measurements in premixed flames using a method of moments*. Fuel, 2018. **222**: p. 287-293.
4. Wu, S., D. Zhou, and W. Yang, *Implementation of an efficient method of moments for treatment of soot formation and oxidation processes in three-dimensional engine simulations*. Applied Energy, 2019. **254**.
5. Mueller, M., G. Blanquart, and H. Pitsch, *Hybrid method of moments for modeling soot formation and growth*. Combustion and Flame, 2009. **156**(6): p. 1143-1155.
6. Wang, Y. and S.H. Chung, *Soot formation in laminar counterflow flames*. Progress in Energy and Combustion Science, 2019. **74**: p. 152-238.
7. E, J., et al., *Soot formation mechanism of modern automobile engines and methods of reducing soot emissions: A review*. Fuel Processing Technology, 2022. **235**: p. 107373.